# Vagus-macrophage-hepatocyte link promotes post-injury liver regeneration and whole-body survival through hepatic FoxM1 activation

Tomohito Izumi[1], Junta Imai[1,2], Junpei Yamamoto[1], Yohei Kawana[1], Akira Endo[1], Hiroto Sugawara[1], Masato Kohata[1], Yoichiro Asai[1], Kei Takahashi[1], Shinjiro Kodama[1], Keizo Kaneko[1], Junhong Gao[1], Kenji Uno[1], Shojiro Sawada[1], Vladimir V. Kalinichenko[3], Yasushi Ishigaki[1,5], Tetsuya Yamada[1] & Hideki Katagiri[1,4]

The liver possesses a high regenerative capacity. Liver regeneration is a compensatory response overcoming disturbances of whole-body homeostasis provoked by organ defects. Here we show that a vagus-macrophage-hepatocyte link regulates acute liver regeneration after liver injury and that this system is critical for promoting survival. Hepatic *Foxm1* is rapidly upregulated after partial hepatectomy (PHx). Hepatic branch vagotomy (HV) suppresses this upregulation and hepatocyte proliferation, thereby increasing mortality. In addition, hepatic FoxM1 supplementation in vagotomized mice reverses the suppression of liver regeneration and blocks the increase in post-PHx mortality. Hepatic macrophage depletion suppresses both post-PHx *Foxm1* upregulation and remnant liver regeneration, and increases mortality. Hepatic *Il-6* rises rapidly after PHx and this is suppressed by HV, muscarinic blockade or resident macrophage depletion. Furthermore, IL-6 neutralization suppresses post-PHx *Foxm1* upregulation and remnant liver regeneration. Collectively, vagal signal-mediated IL-6 production in hepatic macrophages upregulates hepatocyte FoxM1, leading to liver regeneration and assures survival.

---

[1] Department of Metabolism and Diabetes, Tohoku University Graduate School of Medicine, Sendai 980-8575, Japan. [2] Japan Agency for Medical Research and Development, PRIME, Tokyo 100-1004, Japan. [3] Division of Pulmonary Biology, Cincinnati Children's Hospital Medical Center, Cincinnati 45229 Ohio, USA. [4] Japan Agency for Medical Research and Development, CREST, Tokyo 100-1004, Japan. [5]Present address: Division of Diabetes and Metabolism, Department of Internal Medicine, Iwate Medical University, Morioka 020-8505, Japan. Correspondence and requests for materials should be addressed to J.I. (email: imai@med.tohoku.ac.jp)

The liver possesses an enormous capacity for regeneration. After the loss of hepatic tissue caused by toxins or partial surgical removal, remnant hepatocytes markedly proliferate to restore the liver's original mass and functions[1]. This process is a compensatory response overcoming the disturbance of whole-body homeostasis provoked by the functional organ defect. Thus, uncovering the precise mechanism of this regenerative response would lead to better understanding of the whole-body homeostatic system.

While the roles of humoral factors, including cytokines and growth factors, in liver regeneration have been extensively investigated[2], several studies have also suggested a role of vagal nerve signals in liver regeneration[3–5]. After partial hepatectomy (PHx), the liver predominantly regenerates through the proliferation of pre-existing hepatocytes[6]. Hepatocyte proliferation after PHx reportedly begins at 12–16 h and peaks between 36 and 48 h postoperatively in rodents[7]. Reflecting this rapid hepatocyte response, the weight of the remnant liver doubles and has been restored to ~60–70% of the original weight within the first 72 h after 70% PHx, with eventual restoration of the original weight by 14 days after the operation[8,9]. These findings suggest that the liver regenerative process after injury is mechanistically divided into at least two phases, i.e., an early phase characterized by rapid regeneration and a late phase in which the final organ size is determined. In particular, rapid regeneration after severe organ injury leads to prompt functional recovery which is critical for maintaining homeostasis. Hepatic branch vagotomy reportedly suppresses post PHx liver regeneration, especially in the early phase of regeneration[3,4]. However, the mechanisms by which vagal nerve signals induce rapid hepatocyte proliferation after injury remain largely unknown.

Herein, using PHx as an acute organ defect model, we investigated the molecular mechanism underlying acute liver regeneration. Experiments employing blockade of the hepatic branch of the vagus, inducible hepatocyte-specific *FoxM1* deficiency and adenoviral supplementation of hepatic FoxM1 revealed vagal signals to be involved in rapid activation of the hepatocyte FoxM1 pathway, leading to prompt hepatocyte proliferation after PHx. Importantly, while hepatic branch vagotomy increased postoperative mortality, hepatic supplementation of FoxM1 blocked the increase in mortality, indicating the unraveled mechanism to be critical for promoting survival after liver injury. It is noteworthy that macrophages serve as intermediaries for vagal signals and acute activation of the hepatocyte FoxM1 pathway. Acetylcholine, secreted from the vagus, and interleukin (IL)-6, from macrophages, are involved in the molecular mechanism of the vagal nerve–macrophage–hepatocyte transduction. We recently reported vagal signals to directly activate the FoxM1 pathway in pancreatic β-cells, thereby promoting compensatory proliferation[10,11]. In contrast to richly innervated pancreatic islets[12,13], vagal nerves in the liver were very scarce and were observed only around the portal region[14]. Thus, this elaborate multistep mechanism, consisting of neuronal, immune, and parenchymal cells, may allow urgent regenerative signals to be amplified and spread throughout the entire organ, thereby promoting prompt liver regeneration and assuring whole-body survival after severe liver injury.

## Results

**Vagal signals are critical for post PHx survival.** First, we performed 70% PHx concomitantly with hepatic branch vagotomy (HV) (PHx-HV) or sham operation for HV (PHx-sham) in mice, followed by analyzing the survival rates. Notably, the postoperative mortality of PHx-HV mice, within 3 days after surgery, was markedly higher than that in PHx-sham mice or non-

hepatectomized mice subjected to HV alone (HV alone) (Fig. 1a). Histological analysis immediately after their deaths showed that, in contrast to PHx-sham mice, focally necrotic areas with marked bleeding were diffusely located in the livers of PHx-HV mice (Supplementary Fig. 1a). In contrast, regardless of whether or not they had undergone HV, none of the mice, which had survived the initial 3 postoperative days, died during the period of 14 days after the operation (Fig. 1a). In surviving animals, recovery of the hepatic weights of PHx-HV mice was significantly suppressed, as compared with those of PHx-sham mice on day 3 (Supplementary Fig. 1b). Until day 5, the weights of livers from both PHx-sham- and PHx-HV mice were remarkably increased, with an edematous appearance (Supplementary Fig. 1c), and hepatic weights in PHx-HV mice had almost caught up with those in PHx-sham mice on day 7 (Supplementary Fig. 1b). Thus, vagal signals are essential for assuring survival after PHx.

BrdU-positive hepatocytes in PHx-sham mice were remarkably increased as compared with those in mice that had received the sham operation for PHx, by ~30-fold, on day 2 after PHx. In contrast, HV markedly suppressed the increases in BrdU-positive proliferating hepatocytes (Fig. 1b, c). Similar results were obtained when HV was performed 10 days prior to PHx (Supplementary Fig. 1d). Consistent with hepatic weight changes after PHx, the difference in BrdU-positive hepatocyte ratios between PHx-sham- and PHx-HV mice lost significance on days 5 and 7 (Fig. 1b, c). There were no significant differences in body weights or food intakes, during the entire 14-day observation periods after PHx, between surviving PHx-sham mice and PHx-HV mice (Supplementary Fig. 1e, f), indicating that neither HV-induced increments in post-injury mortality nor suppression of the liver regenerative response was attributable mainly to general conditions, including nutritional states. These findings suggest that vagal signals play important roles in acute-phase liver regeneration, which appears to be critical for the post PHx survival.

**Vagal signals are involved in activation of hepatic FoxM1 after PHx.** We next investigated the molecular mechanism underlying the vagal signal-mediated acute liver regeneration after PHx. We recently reported that vagal signals promote pancreatic β-cell proliferation via an intracellular FoxM1-dependent mechanism[11]. FoxM1 affects several aspects of cell cycle progression[15], such as promoting the G1/S transition by triggering the transcriptions of several cyclins, including cyclin A2 (*Ccna2*)[16–18], regulation of the G2/M transition by transactivating cyclin-dependent kinase 1 (*Cdk1*)[17,19], and control of proper progression of mitosis by increasing the expressions of several mitogenic genes, such as polo-like kinase 1 (*Plk1*)[20]. Therefore, we explored whether FoxM1 expression in hepatocytes is involved in vagal signal-mediated cellular proliferation. PHx markedly upregulated *Foxm1* and cell cycle-related genes downstream from FoxM1, such as *Cdk1*, *Ccna2,* and *Plk1*, as well as a cell proliferation marker, *Mki67*, by several dozen to 100-fold on postoperative days 2 and 3 (Fig. 2a and Supplementary Fig. 1g). Importantly, PHx-induced upregulations of *FoxM1* and its related genes were almost completely blocked by HV (Fig. 2a and Supplementary Fig. 1g). Consistently, immunoblotting revealed that expressions of hepatic FoxM1 and its downstream proteins, such as Cyclin A2, Cdk1, and PLK1, were markedly increased on postoperative day 2 and that these increases were blocked by HV (Fig. 2b). Thus, vagal signals are essential for activation of the hepatic FoxM1 pathway in the early phase after PHx, leading to acute hepatocyte replication. Again, in contrast to the early postoperative periods, HV had much weaker and minimal, respectively, impacts on PHx-induced upregulations of FoxM1-related gene expressions

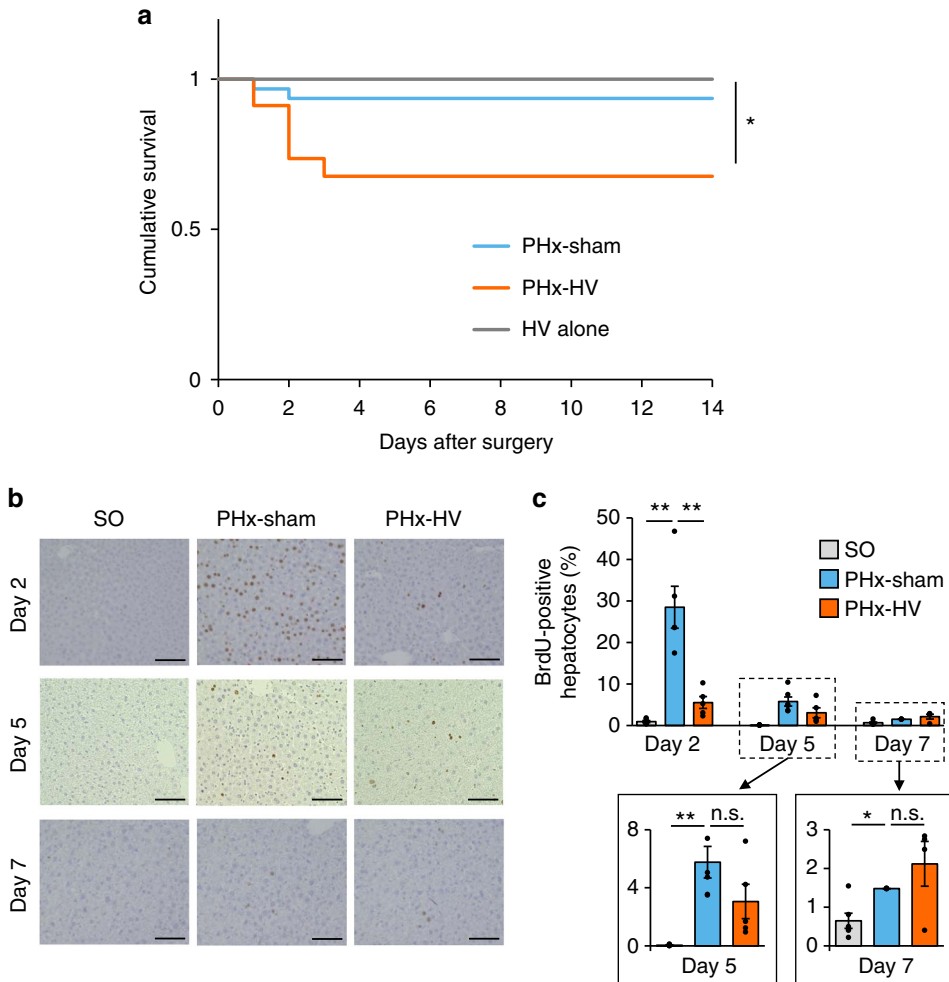

**Fig. 1** Vagal signals are critical for post PHx survival. **a** Cumulative survival of mice that underwent PHx-sham ($n = 26$), PHx-HV ($n = 26$), and HV alone ($n = 18$). **b** Representative images of liver sections immunostained for BrdU on days 2, 5, and 7 after sham operation for PHx (SO), PHx-sham, or PHx-HV. Scale bars indicate 100 μm. **c** BrdU-positive hepatocyte ratios in SO mice ($n = 5$–6 per group), PHx-sham mice ($n = 4$–6 per group), and PHx-HV mice ($n = 4$–6 per group) on postoperative days 2, 5, and 7. The magnified graphs on days 5 and 7 are shown in framed boxes with lowered scale ranges. *$P < 0.05$; **$P < 0.01$ assessed by log-rank test with Bonferroni correction (**a**) or one-way ANOVA followed by Bonferroni's post hoc test (**c**). n.s., not significant

on days 5 and 7 (Supplementary Fig. 1h, i), although the magnitudes of increases differed due to using different detection procedures. Thus, hepatocyte regeneration in the late phase is unlikely to be mediated by vagal signals.

**FoxM1 activation promotes liver regeneration and survival**. To explore whether the FoxM1 upregulation observed in the early phase after PHx is actually responsible for post PHx liver regeneration, we generated tamoxifen-inducible liver-specific FoxM1 knockout mice (iFoxM1LKO mice) by crossing serum albumin promoter-CreER mice[21] and FoxM1-floxed mice[22]. In contrast to congenital liver-specific FoxM1 knockout mice[22], iFoxM1LKO mice are not deficient in hepatocyte FoxM1 during developmental and growth stages. Tamoxifen administration to these mice markedly decreased *Foxm1* expressions in the liver, by 71% (Supplementary Fig. 2). As expected, PHx-induced upregulations of the cell cycle-related genes (Fig. 3a) and increases in BrdU-positive hepatocytes (Fig. 3b) on day 2 after the surgeries were markedly blunted in iFoxM1LKO mice, showing that FoxM1 expressed in hepatocytes is essential for accelerating the hepatocyte cell cycle, thereby promoting hepatocyte replication in the early phase after PHx.

Next, to explore the role of FoxM1 in vagal signal-mediated liver regeneration, we examined the effects of HV on liver regeneration after PHx using mice, in which hepatic FoxM1 had been supplemented by adenoviral gene transduction. We used adenovirus containing the human *Foxm1* gene to distinguish endogenous FoxM1 from exogenously expressed FoxM1. We selected an adenoviral titer at which gene transduction increased exogenous *Foxm1* expression in the livers of mice to levels similar to those of hepatic endogenous FoxM1 in PHx-sham mice induced on day 2 after PHx (Supplementary Fig. 3a). Strikingly, in hepatically FoxM1-supplemented mice, HV failed to suppress PHx-induced increments in expressions of the genes downstream from FoxM1 (Fig. 3c). This effect was also observed in BrdU-positive hepatocytes (Fig. 3d). Notably, hepatic FoxM1 supplementation completely blocked increases in mortality after PHx in vagotomized mice (Fig. 3e). Interestingly, exogenous (human) FoxM1 expression in normal mice, without PHx or HV, did not alter the expressions of either endogenous mouse *FoxM1* or its target genes, nor that of *Mki67* (Supplementary Fig. 3b). Furthermore, hepatic FoxM1 overexpression in PHx-sham mice did not yield further enhancements of hepatocyte proliferative responses, suggesting that the hepatic FoxM1

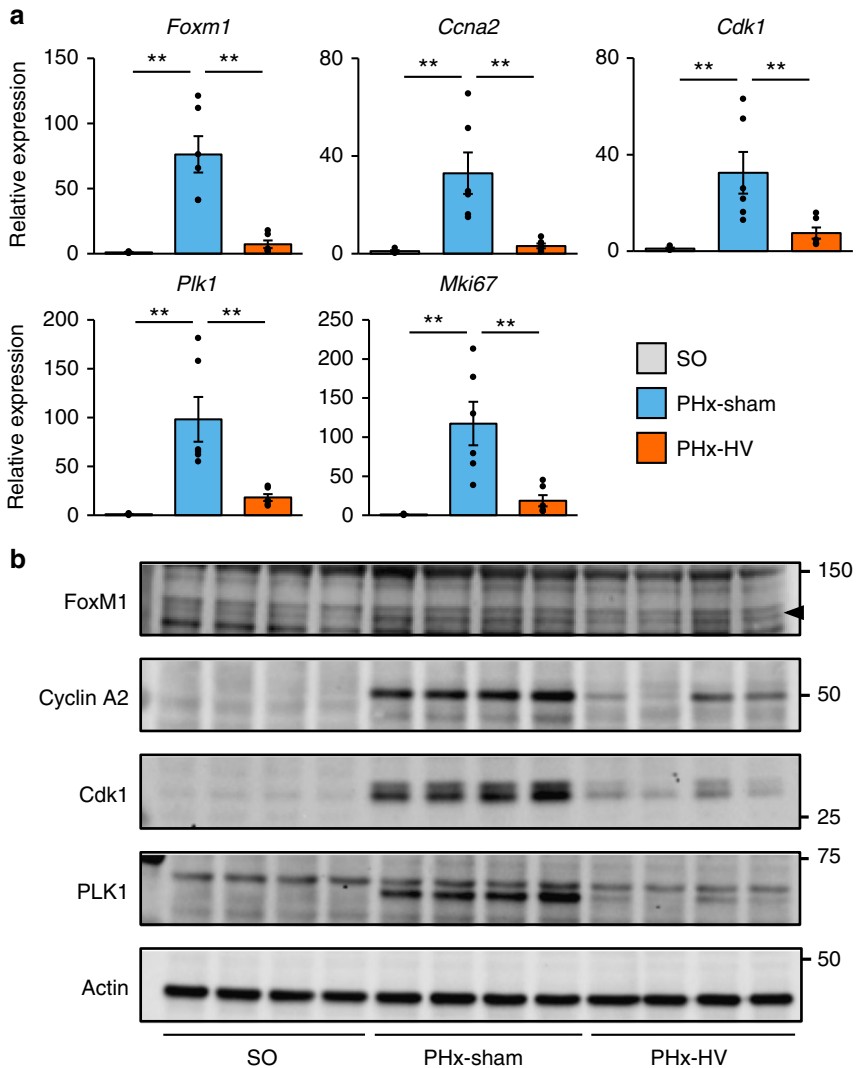

**Fig. 2** Vagal signals are involved in activation of hepatic FoxM1 after PHx. **a** Relative expressions of *Foxm1*, its target genes. and *MKi67* in the liver 2 days after SO (*n* = 4), PHx-sham (*n* = 6), and PHx-HV (*n* = 6). **b** Images of liver extract immunoblottings with anti-FoxM1, Cyclin A2, Cdk1, PLK1, and actin in SO, PHx-sham, and PHx-HV mice on postoperative day 2. Arrowhead indicates bands for FoxM1. Data are presented as means ± SEM. *$P < 0.01$ assessed by one-way ANOVA followed by Bonferroni's post hoc test (**a**). n.s., not significant

pathway had already been fully activated in response to PHx (Supplementary Fig. 3c, d). In addition, administration of carbachol to normal mice failed to upregulate hepatic FoxM1-related genes (Supplementary Fig. 3e). These results indicate that, under early-phase post-PHx conditions, activation of the hepatic FoxM1 pathway mediates vagal signal-induced promotion of hepatocyte replication and the resultant whole-body survival.

**Macrophages mediate vagus-induced hepatic FoxM1 activation.** Our next goal was to elucidate the mechanism by which vagal signals activate the FoxM1 pathway in hepatocytes. First, to examine whether the muscarinic receptor signaling induced by acetylcholine, the main neurotransmitter released by the vagal nerve, is involved, mice were treated with atropine, a muscarinic receptor antagonist, followed by PHx. With atropine treatment, the increases in BrdU-positive hepatocytes were markedly reduced (Fig. 4a) and recovery of hepatic weights in the early phase after PHx was suppressed (Fig. 4b). Consistently, upregulations of hepatic *Foxm1* and its target genes, as well as *Mki67* after PHx were significantly blocked by atropine treatment

(Fig. 4c). These data indicate that muscarinic signals are involved in promoting acute liver regenerative responses after PHx.

We then attempted to elucidate the molecular mechanism underlying vagal signal-mediated hepatocyte proliferation in vitro by using a murine hepatoma cell line, Hepa1–6 cells. Unexpectedly, however, direct stimulation with carbachol, a cholinergic receptor agonist, did not upregulate *Foxm1* and *Mki67* gene expressions in the cultured hepatocytes (Supplementary Fig. 4a). Therefore, we examined the possibility that an indirect mechanism(s) mediates the effects of vagal signals on hepatocyte proliferation. Since resident macrophages are reportedly involved in tissue repair in several tissues[23], we examined post PHx hepatocyte proliferation using mice in which hepatic macrophages had been depleted by treatment with clodronate liposomes. Clodronate liposome administration significantly decreased hepatic expressions of macrophage markers (Fig. 5a), indicating successful depletion of hepatic macrophages. Intriguingly, clodronate liposome administration markedly blunted activation of the FoxM1 pathway in the liver on day 2 after PHx, as compared with control liposome treatment (Fig. 5b). In addition, the increases in BrdU-positive hepatocytes after PHx

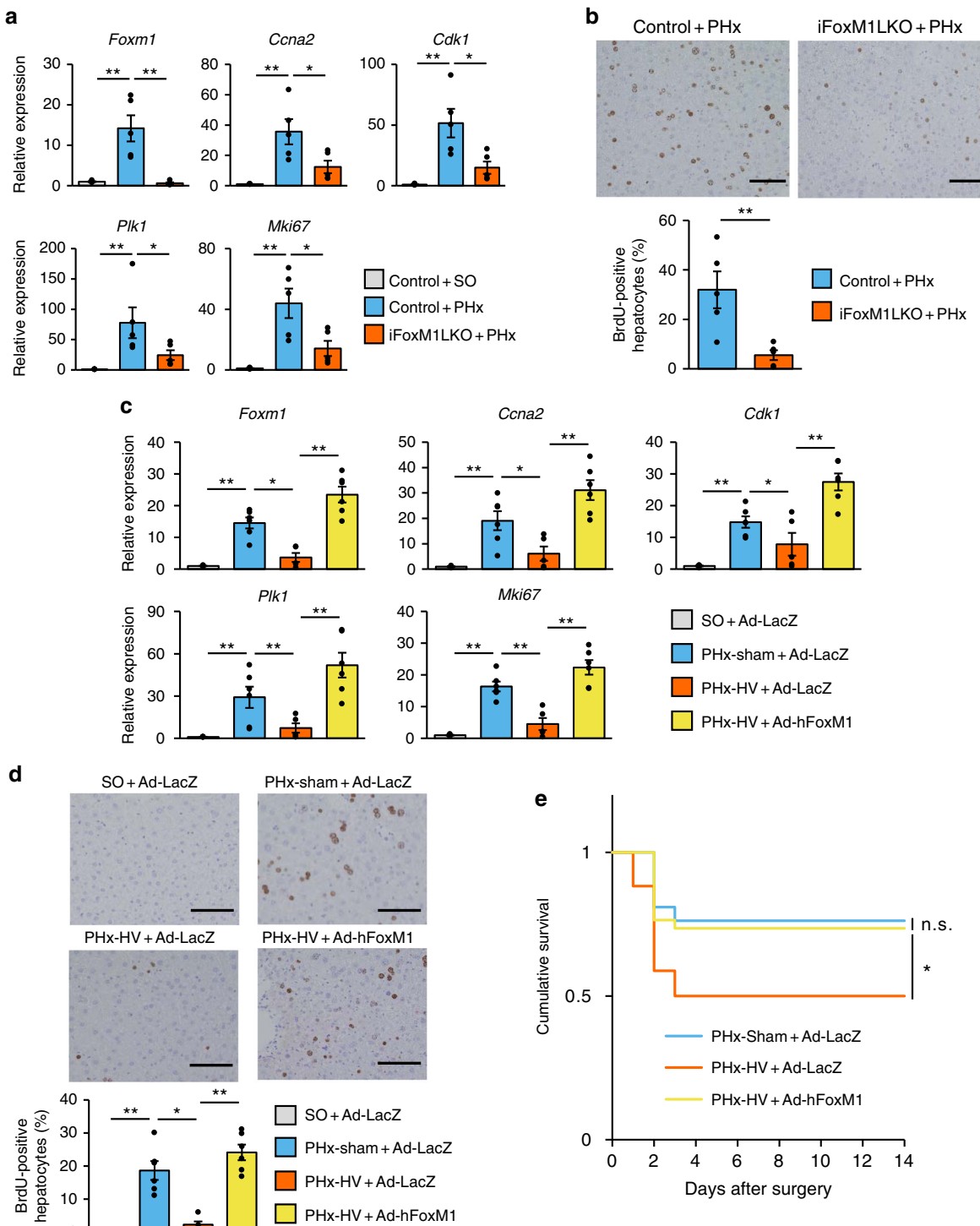

**Fig. 3** FoxM1 activation promotes liver regeneration and whole-body survival. **a** Relative expressions of *Foxm1*, its target genes, and *MKi67* in the liver 2 days after surgery in control mice that underwent SO (*n* = 5) and PHx (*n* = 5), and iFoxM1LKO mice that underwent PHx (*n* = 5). **b** (Upper panels) Representative images of liver sections immunostained for BrdU 2 days after PHx from control and iFoxM1LKO mice. Scale bars indicate 100 μm. (Lower panel) BrdU-positive hepatocyte ratios in control (*n* = 5) and iFoxM1LKO (*n* = 5) mice 2 days after PHx. **c** Relative expressions of *Foxm1*, its target genes, and *MKi67* in the liver 2 days after SO (*n* = 5), PHx-sham (*n* = 6), and PHx-HV (*n* = 5) in mice receiving Ad-LacZ and after PHx-HV in mice receiving Ad-hFoxM1 (*n* = 6). **d** (Upper panels) Representative images of liver sections immunostained for BrdU 2 days after SO, PHx-sham, and PHx-HV from mice receiving Ad-LacZ and after PHx-HV from mice receiving Ad-hFoxM1. Scale bars indicate 100 μm. (Lower panel) BrdU-positive hepatocyte ratios in the liver 2 days after SO (*n* = 5), PHx-sham (*n* = 6), and PHx-HV (*n* = 5) in Ad-LacZ-treated mice and PHx-HV in Ad-hFoxM1-treated mice (n = 6). **e** Cumulative survival of mice that underwent PHx-sham (*n* = 21) and PHx-HV (*n* = 34) procedures after receiving Ad-LacZ and mice that underwent PHx-HV after receiving Ad-hFoxM1 (*n* = 34). Data are presented as means ± SEM. *$P < 0.05$; **$P < 0.01$ assessed by one-way ANOVA followed by Bonferroni's post hoc test (**a**, **c**, **d**), unpaired *t* test (**b**), or log-rank test with Bonferroni correction (**e**). n.s., not significant

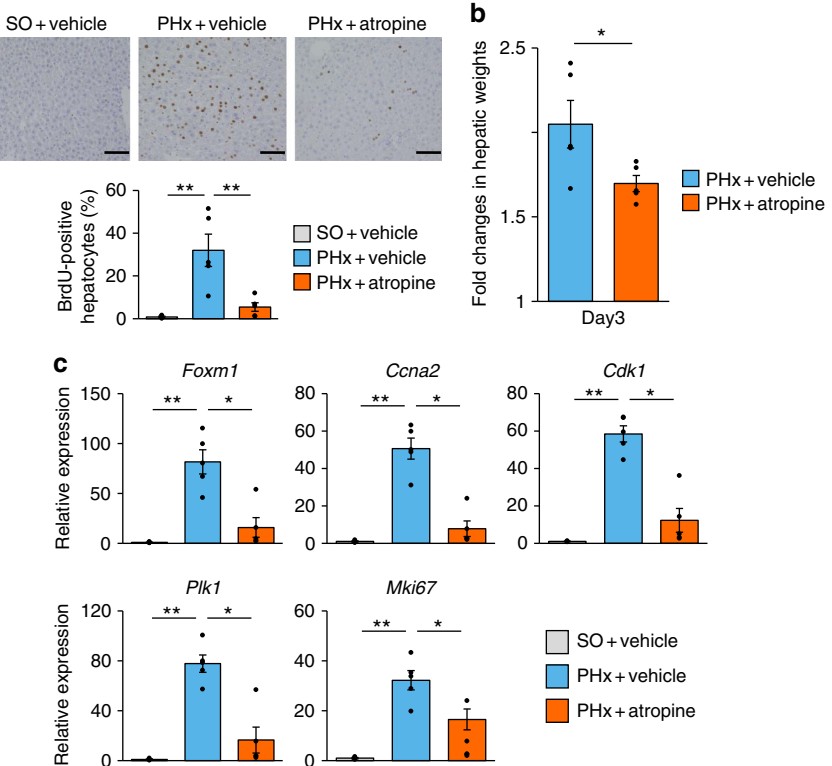

**Fig. 4** Muscarinic signals are involved in acute liver regenerative responses after PHx. **a** (Upper panels) Representative images of liver sections immunostained for BrdU on day 2 after sham operation for PHx (SO) and PHx in vehicle-treated mice and after PHx in atropine-treated mice. Scale bars indicate 100 μm. (Lower panel) BrdU-positive hepatocyte ratios in SO + vehicle mice ($n = 4$), PHx + vehicle mice ($n = 5$), and PHx + atropine mice ($n = 5$). **b** Fold changes in hepatic weights 3 days after surgery in PHx + vehicle mice ($n = 5$) and PHx + atropine mice ($n = 5$). Hepatic weights were divided by those obtained immediately after surgery. **c** Relative expressions of *Foxm1*, its target genes, and *MKi67* in the liver 2 days after SO ($n = 4$) and PHx ($n = 5$) in vehicle-treated mice and PHx ($n = 5$) in atropine-treated mice. *$P < 0.05$; **$P < 0.01$ assessed by one-way ANOVA followed by Bonferroni's post hoc test (**a** and **c**) or assessed by unpaired $t$ test (**b**). n.s., not significant

were markedly inhibited by clodronate liposome administration (Fig. 5c). Of note, HV failed to exert further inhibitory effects on the acute hepatocyte proliferation responses of clodronate liposome-treated mice (Fig. 5b, c). Furthermore, adenoviral supplementation of hepatic FoxM1 significantly blunted the inhibitory effects of macrophage depletion on acute hepatocyte proliferation responses after PHx (Fig. 5d, e). Notably, the post-PHx mortality of macrophage-depleted mice, within 2 days postoperatively, was markedly higher than those of mice without macrophage depletion (Fig. 5f), observations similar to those in mice undergoing HV (Fig. 1a). These findings, taken together, suggest that hepatic macrophages contribute to mediating vagal signals to hepatocytes, thereby activating the hepatocyte FoxM1 pathway and promoting hepatocyte proliferation.

**Vagal signals increase macrophage IL-6 production after PHx.**
What then is the mediator of the macrophage–hepatocyte link? Activated macrophages are well-known to produce and release pro-inflammatory cytokines, including IL-6[24], and the cellular source of IL-6 after PHx was reported to be resident macrophages[25]. Therefore, we focused on IL-6 production after PHx. *Il-6* expression in the remnant liver of PHx-sham mice was markedly increased at 6 h after PHx and, importantly, this *Il-6* upregulation was almost completely abolished by HV (Fig. 6a). Plasma IL-6 concentrations were also increased after PHx, and these increases were suppressed by vagotomy (Supplementary Fig. 4b). In addition, upregulation of *Il-6* by PHx was completely absent in clodronate liposome-treated mice (Fig. 4b). Consistent with the

results of hepatocyte proliferative responses (Fig. 5b, c), HV failed to exert further inhibitory effects on upregulation of *Il-6* in clodronate liposome-treated mice (Fig. 6b). Therefore, these findings indicate that, after PHx, vagal signals enhance IL-6 production by hepatic macrophages in the remnant liver.

We next investigated the effects of cholinergic signals on macrophage IL-6 production employing ex vivo experiments using primary macrophages isolated from the peritoneal cavity after intraperitoneal thioglycollate administration. Strikingly, carbachol treatment markedly increased *Il-6* expression in these primary macrophages in a dose-dependent manner (Supplementary Fig. 4c), and this increase was significantly blunted by co-treatment with atropine (Fig. 6c). In contrast, carbachol treatment of primary hepatocytes failed to increase *Il-6* expression (Supplementary Fig. 4d). Moreover, in vivo atropine treatment significantly suppressed PHx-induced increases in *Il-6* expression in the liver (Fig. 6d). Collectively, vagus-derived cholinergic signals can directly upregulate macrophage IL-6 production through a muscarinic receptor-dependent mechanism.

**IL-6 mediates vagal signal-induced hepatic FoxM1 activation.**
To examine whether IL-6 activates the FoxM1 pathway in hepatocytes, we treated primary hepatocytes with IL-6. Baseline *Foxm1* levels in primary hepatocytes were approximately eight-fold higher than those in non-treated quiescent livers (Supplementary Fig. 5a). The isolation procedure may intrinsically enhance *Foxm1* expression. Even under these conditions, IL-6 treatment of primary hepatocytes further increased expressions of

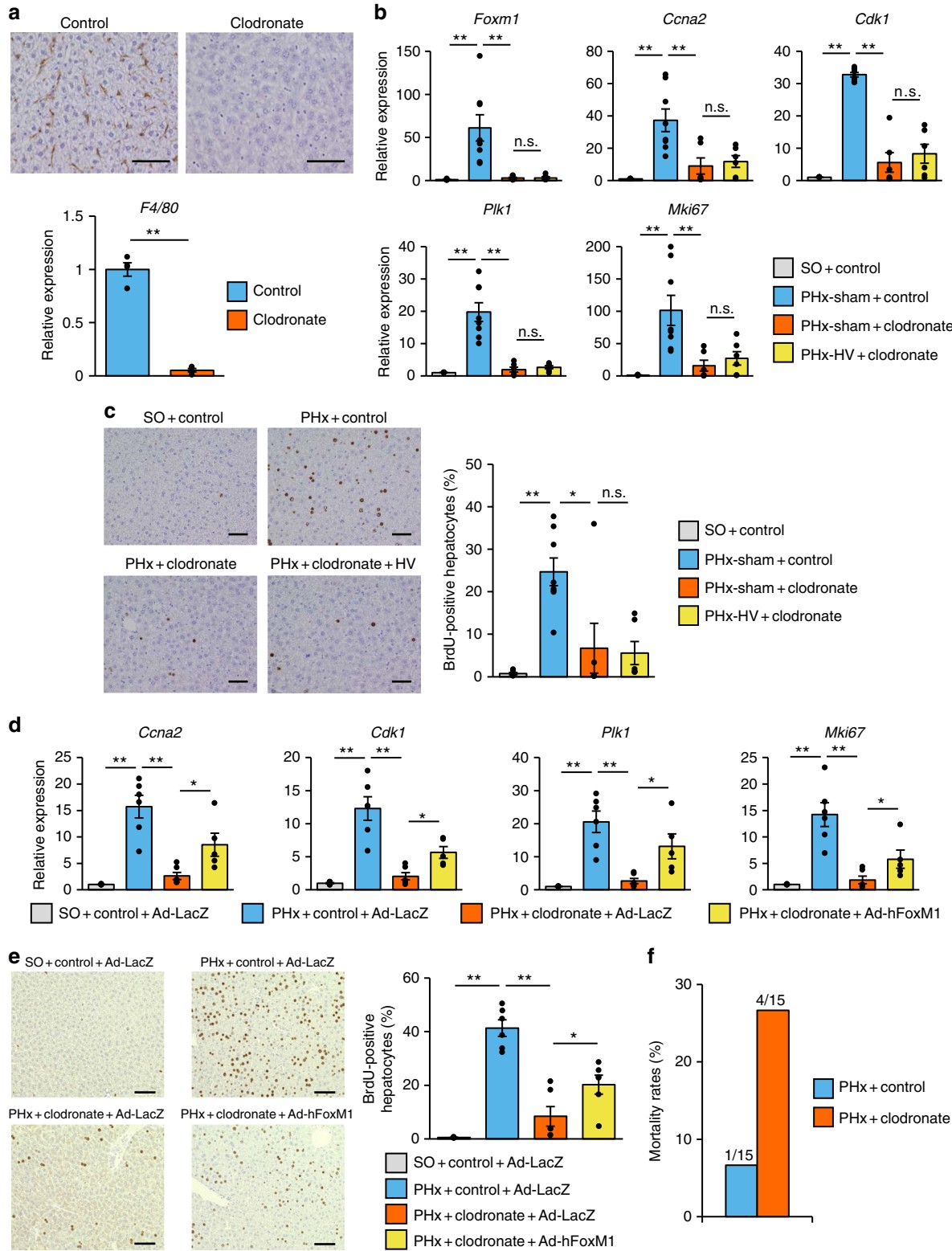

*Foxm1*, its target genes, and *Mki67* (Fig. 7a). It should be noted that IL-6-induced upregulations of these cell cycle-related genes were markedly blocked in primary hepatocytes from iFoxM1LKO mice (Fig. 7a). These findings indicate that IL-6 accelerates cell cycling of hepatocytes through a FoxM1-dependent mechanism.

IL-6 is well-known to exert its effects through signal transducer and activator of transcription 3 (STAT3) signaling[26]. IL-6 promotes phosphorylation and homodimerization of STAT3,

thereby increasing STAT3 translocation to the nucleus, leading to transactivation of its target genes[26]. Since STAT3 reportedly transactivates *Foxm1* gene expression[27], we next examined hepatic STAT3 phosphorylation of PHx-sham- and PHx-HV mice. Compared with control mice, hepatic STAT3 phosphorylation was significantly increased at 24 h after PHx, and this enhancement of phosphorylation was blunted by HV (Fig. 7b). Next, we pretreated primary hepatocytes with STAT3 inhibitor

**Fig. 5** Macrophages mediate vagus-induced hepatic FoxM1 activation. **a** (Upper panels) Representative images of liver sections immunostained for F4/80 from mice 24 h after receiving control and clodronate liposomes. Scale bars indicate 100 μm. (Lower panel) Relative gene expressions of *F4/80* in the liver 24 h after control (*n* = 4) and clodronate (*n* = 4) liposome administration. **b** Relative expressions of *Foxm1*, its target genes, and *MKi67* in the liver 2 days after surgery in SO + control mice (*n* = 6), PHx-sham + control mice (*n* = 8), PHx-sham + clodronate mice (*n* = 6), and PHx-HV + clodronate mice (*n* = 6). **c** (Left panels) Representative images of liver sections immunostained for BrdU on day 2 in SO + control-, PHx-sham + control-, PHx-sham + clordronate-, and PHx-HV + clodronate mice. Scale bars indicate 100 μm. (Right panel) BrdU-positive hepatocyte ratios in SO + control mice (*n* = 6), PHx-sham + control mice (*n* = 8), PHx-sham + clodronate mice (*n* = 6,) and PHx-HV + clodronate mice (*n* = 6) 2 days after surgery. **d** Relative expressions of *Foxm1*, its target genes, and *MKi67* in the liver 2 days after SO in mice receiving control liposome and Ad-LacZ (*n* = 6) and PHx in mice receiving control liposome and Ad-LacZ (*n* = 6), clodronate liposome and Ad-LacZ (*n* = 6), and clodronate liposome and Ad-hFoxM1 (*n* = 6). **e** (Left panels) Representative images of liver sections immunostained for BrdU on day 2 after SO in mice receiving control liposome and Ad-LacZ and PHx in mice receiving control liposome and Ad-LacZ, clodronate liposome and Ad-LacZ, and clodronate liposome and Ad-hFoxM1. Scale bars indicate 100 μm. (Right panel) BrdU-positive hepatocyte ratios in the liver 2 days after SO in mice receiving control liposome and Ad-LacZ (*n* = 6) and PHx in mice receiving control liposome and Ad-LacZ (*n* = 6), clodronate liposome and Ad-LacZ (*n* = 6), and clodronate liposome and Ad-hFoxM1 (*n* = 6). **f** Mortality rates within 2 postoperative days in PHx + control mice (*n* = 15) and PHx + clodronate mice (*n* = 15). Data are presented as means ± SEM. *P < 0.05; **P < 0.01 assessed by one-way ANOVA followed by Bonferroni's post hoc test (**b**, **c**, **d**, **e**) or assessed by unpaired *t* test (**a**). n.s., not significant

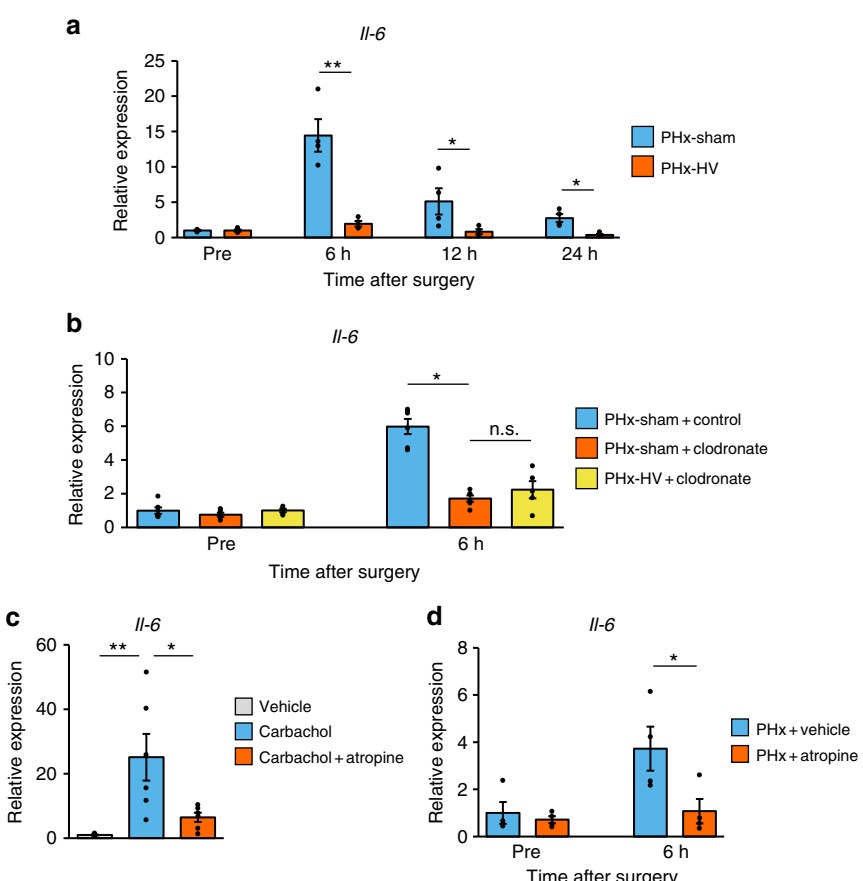

**Fig. 6** Vagal signals increase macrophage IL-6 production after PHx. **a** Relative gene expressions of *Il-6* after surgery in PHx-sham mice (*n* = 4 per group) and PHx-HV mice (*n* = 4 per group). **b** Relative gene expressions of *Il-6* after PHx-sham in control liposome-treated (*n* = 5–6 per group) and clodronate liposome-treated mice (*n* = 5–6 per group), and after PHx-HV in clodronate liposome-treated mice (*n* = 5–6 per group). **c** Relative gene expression of *Il-6* in primary macrophages after 4 h of stimulation with vehicle (*n* = 6), 100 μM carbachol (*n* = 6), or both 100 μM carbachol and 100 μM atropine (*n* = 6). **d** Relative gene expressions of *Il-6* after PHx in vehicle-treated (*n* = 4 per group) and atropine-treated (*n* = 4 per group) mice. *P < 0.05; **P < 0.01 assessed by unpaired *t* test (**a** and **d**), or assessed by one-way ANOVA followed by Bonferroni's post hoc test (**b** and **c**). n.s., not significant

peptide, which inhibits both homodimerization of STAT3 and heterodimerization of STAT3 and STAT1[28], followed by treatment with IL-6. Pretreatment with the STAT3 inhibitor blocked IL-6-mediated increases in *FoxM1* and its target genes as well as *Mki67* (Supplementary Fig. 5b). In addition, we examined the phosphorylation of STAT1 as well as that of STAT5 after PHx. In contrast to STAT3, phosphorylations of STAT1 and STAT5 were

not altered after PHx (Supplementary Fig. 5c). These results suggest that IL-6 promotes phosphorylation and homodimerization of STAT3, thereby upregulating FoxM1-related genes in hepatocytes.

Finally, to examine the significance of IL-6 in activation of the hepatic STAT3–FoxM1 pathway in vivo, we blocked IL-6 signaling by administering anti-IL-6 antibody, followed by PHx

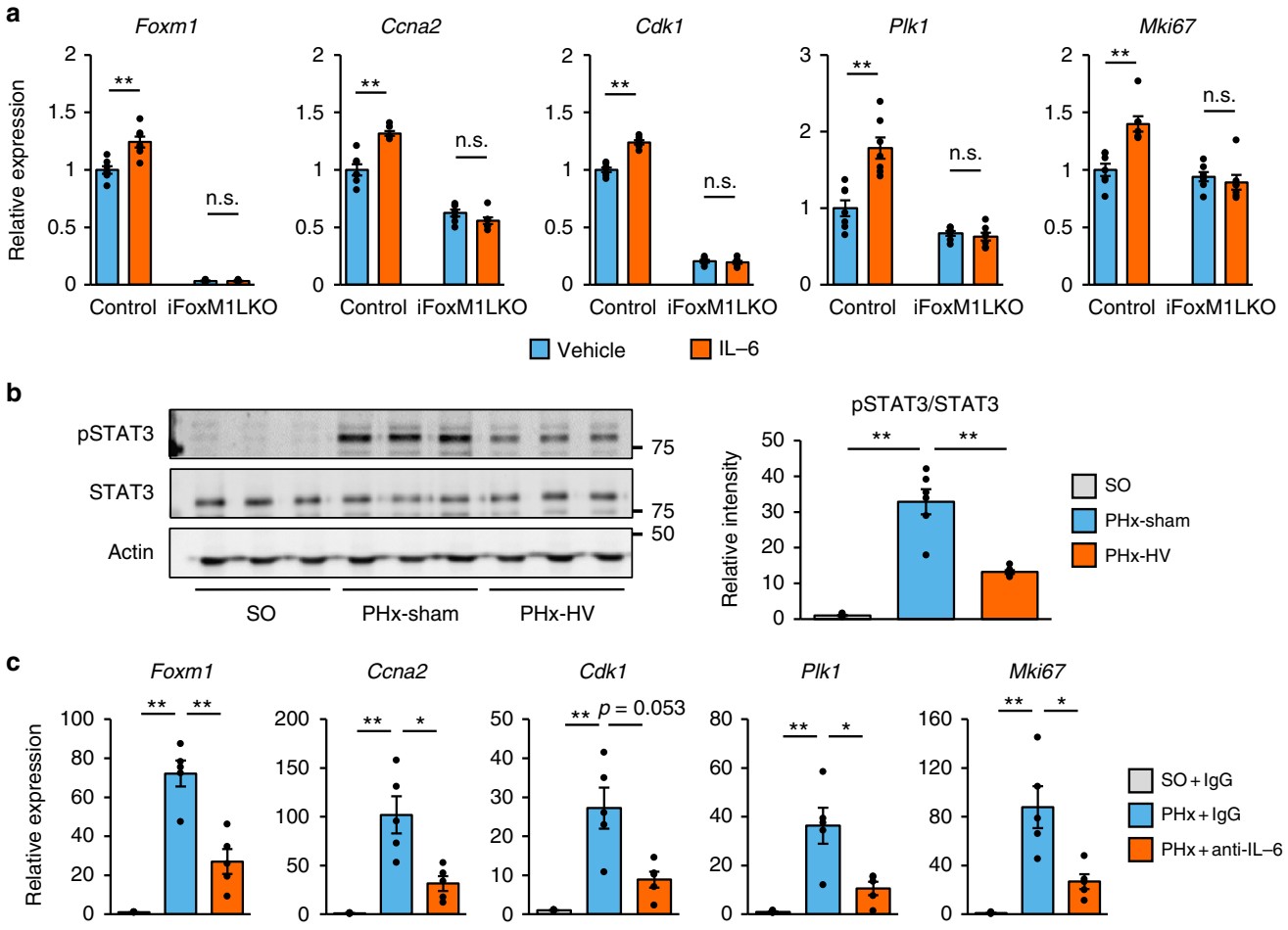

**Fig. 7** IL-6 mediates vagal signal-induced hepatic FoxM1 activation. **a** Relative expressions of *Foxm1*, its target genes, and *MKi67* in primary hepatocytes isolated from control and iFoxM1LKO mice treated with 100 ng/ml IL-6 (*n* = 7–8 per group) and vehicle (*n* = 7–8 per group) for 6 h. **b** (Left panels) Representative images of liver extract immunoblottings with anti-phospho-STAT3, total STAT3, and actin. (Right panel) Relative intensities of phospho/ total STAT3 in livers from sham operation for PHx (SO)- (*n* = 6), PHx-sham- (*n* = 6), and PHx-HV mice (*n* = 6). **c** Relative expressions of *Foxm1*, its target genes, and *MKi67* in the liver 2 days after surgery from SO + IgG (*n* = 5), PHx + IgG (*n* = 5), and PHx + anti-IL-6 antibody- (*n* = 5) treated mice. Data are presented as means ± SEM. *$P$ < 0.05; **$P$ < 0.01 assessed by unpaired *t* test (**a**), or assessed by one-way ANOVA followed by Bonferroni's post hoc test (**b** and **c**). n.s., not significant

surgery. IL-6 neutralization significantly suppressed the hepatic STAT3 phosphorylation enhanced by PHx (Supplementary Fig. 5d). Under these conditions, upregulations of *Foxm1* and its target cell cycle-related genes, as well as *Mki67* on day 2 after PHx were markedly suppressed in anti-IL-6 antibody-treated mice (Fig. 7c). Thus, IL-6 signaling plays key roles in activation of the FoxM1 pathway in hepatocytes and in the proliferation of hepatocytes during the acute phases after PHx.

Taken together, our observations suggest that vagal nerve-derived cholinergic signals are likely to stimulate IL-6 production by resident macrophages in the liver, and that, in a paracrine manner, IL-6 enhances hepatocyte proliferation based on a FoxM1-dependent mechanism (Fig. 8).

## Discussion

Using the PHx model, we have clarified that vagal signal-regulated acute liver regenerative responses are critical for promoting survival after severe liver injury. Postoperative mortality was high in PHx-HV mice but, regardless of whether or not HV had been performed, all of the mice which had survived the initial 3 postoperative days survived for at least 14 days after the

operation (Fig. 1a). Thus, acute recovery of liver mass within a few days after PHx is critical for life maintenance. Increases in postoperative mortality in PHx-HV mice were accompanied by areas of focal necrosis with marked bleeding in the livers of these mice. Further studies are necessary to elucidate the mechanisms by which HV leads to such pathological changes. Meanwhile, hepatic FoxM1 supplementation in PHx-HV mice completely blocked the increases in mortality (Fig. 3e) with recovery of acute liver regeneration (Fig. 3c, d). On the other hand, adenoviral FoxM1 expression or carbachol administration to normal mice had no effects on the FoxM1 pathway (Supplementary Fig. 3b, e). These findings suggest that additional and unknown signals, induced by PHx, are necessary for promoting hepatocyte proliferation. Thus, vagus-mediated activation of the FoxM1 pathway is critical, especially when prompt hepatocyte regeneration is required, and functions as a preventive mechanism against whole-body death after injury. In contrast, chronic liver regeneration occurred even in PHx-HV mice, indicating different mechanisms and roles of liver regeneration in the acute versus chronic phases. Late-phase liver regeneration was proposed to be promoted by humoral factors including several growth factors[7], and could be important in regulating liver size. On the other hand, neuronal

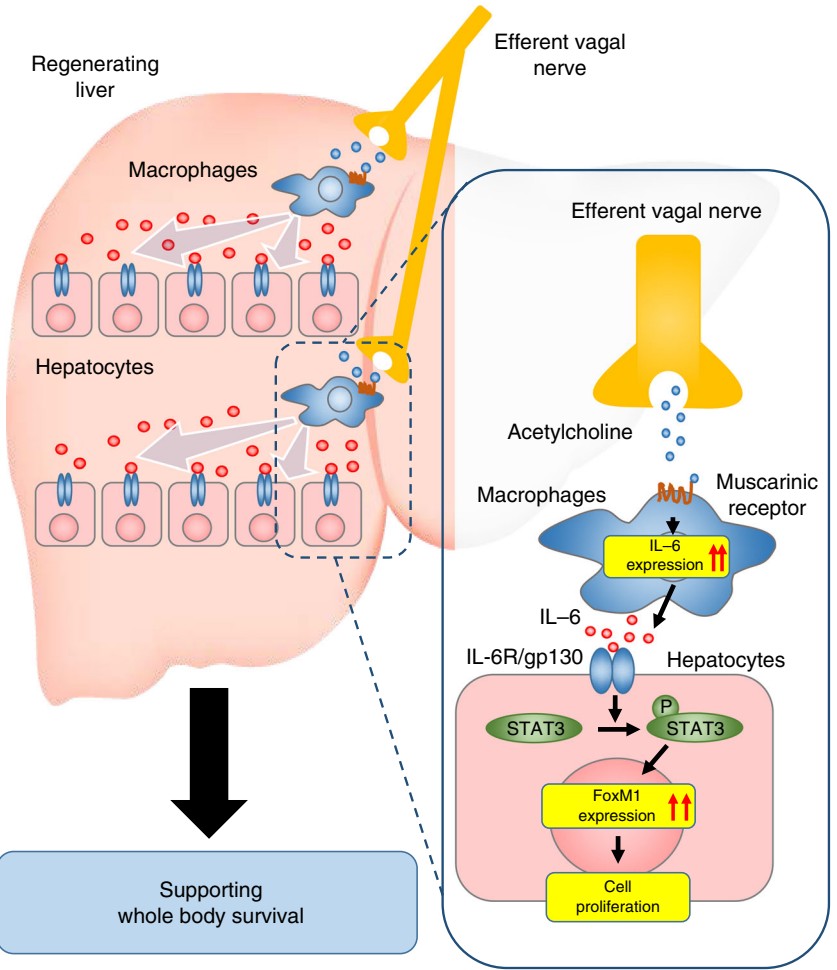

**Fig. 8** The proposed mechanism of vagus-mediated liver regeneration. The multistep regulatory mechanism of acute liver regeneration after PHx via a vagus–macrophage–hepatocyte link

signals may be advantageous for inducing prompt compensatory regeneration, which confers a survival advantage for animals suffering severe crises such as an organ defect. Recovery of liver weight within 72 h after PHx is reportedly reduced in 40- as compared to 4-week-old mice[29]. It is noteworthy that 40-week-old mice showed a low survival rate within 72 h after PHx[29], similar to that of PHx-HV mice in the present study. Taken together with our results, the lower survival rate after PHx associated with aging might be attributable to impairment of the mechanism, underlying acute liver regenerative responses, elucidated in the present study.

In this study, we have delineated a full picture of vagal signal-regulated acute hepatic regenerative responses. We have unmasked several novel mechanisms involved in this process. First, vagal nerve signals play critical roles in upregulating the hepatocyte FoxM1 pathway, thereby promoting rapid liver regeneration in the early phases after injury. Second, resident macrophages mediate processes involving vagal signals and prompt hepatocyte proliferation in these regenerative responses. Third, IL-6 is a critical mediator in the mechanism linking macrophages and hepatocytes.

Consistent with the results of a previous study using congenital FoxM1 knockout mice[22], liver-specific induction of FoxM1 deficiency prior to PHx markedly suppressed liver regeneration in the early phase after PHx (Fig. 3a, b). Notably, we further showed that HV did not suppress liver regeneration and whole-body survival in hepatically FoxM1-supplemented mice (Fig. 3d, e).

Thus, vagal signal-induced FoxM1 upregulation in hepatocytes is necessary and sufficient for accelerating the hepatocyte cell cycle, thereby promoting hepatocyte replication in the early phase after PHx. Given that vagus-mediated FoxM1 activation is involved in compensatory β-cell proliferation[11], the vagus-FoxM1 linking mechanism may be shared by a variety of tissues for in vivo regeneration.

Direct carbachol stimulation did not upregulate FoxM1-related genes in hepatocytes (Supplementary Fig. 4a), but did markedly enhance IL-6 production in macrophages (Fig. 6c), suggesting the targets of vagal signals to be hepatic macrophages, rather than hepatocytes. As reported[24,30], hepatic macrophage-depleted mice showed marked impairment of liver regeneration (Fig. 5c). Importantly, this impairment was accompanied by blunting of FoxM1 pathway activation (Fig. 5b), and vagotomy failed to exert further inhibitory effects on either acute liver regenerative responses (Fig. 5b, c) or the upregulation of hepatic *Il-6* (Fig. 6b). Vagal nerve signals reportedly regulate resident macrophages, thereby modulating several biological conditions in peripheral organs, such as inflammation[31,32] and metabolism[33]. In this study, we clarified that the vagal nerve–macrophage link is further involved in the organ regenerative process. In this process, the cholinergic signal is identified as a stimulator of hepatic macrophages. Cholinergic signals reportedly promote anti-inflammatory responses of macrophages through the α7-nicotinic receptor[31,32]. In addition, activation of muscarinic signals in the central nervous system has systemic anti-inflammatory

properties, and these effects are reportedly elicited by efferent vagal nerve-mediated nicotinic[34], rather than muscarinic, actions[35] on peripheral immune cells. Meanwhile, since atropine treatment blunted the carbachol-mediated (Fig. 6c) and PHx-induced (Fig. 6d) upregulations of IL-6 in macrophages ex vivo and in the remnant liver in vivo, respectively, muscarinic signaling is involved in liver regeneration. Intriguingly, inflammatory responses opposite those of hepatic macrophages to cholinergic signals, depending on whether the signals are mediated by muscarinic or nicotinic receptors, were recently reported[33]. Thus, responses of macrophages to vagal signals are likely to differ depending on physiological or pathological situations. Expression levels of muscarinic and nicotinic receptors on macrophages might be altered according to various situations, although further examinations are needed to elucidate the molecular mechanism underlying the opposite responses of macrophages to cholinergic stimulation.

In terms of the macrophage–hepatocyte link, IL-6 was revealed to be a key mediator. Il-6 expression in the liver was rapidly and markedly increased after PHx, and this increase was found to be suppressed by vagotomy (Fig. 6a) and in hepatic macrophage-depleted mice (Fig. 6b). IL-6 treatment activated the FoxM1 pathway in hepatocytes ex vivo (Fig. 7a), and activation of the FoxM1 pathway after PHx was inhibited by antibody-mediated blockade of hepatic IL-6 signaling in vivo (Fig. 7c). In addition, PHx enhanced phosphorylation of hepatic STAT3 (Fig. 7b), a downstream target of the IL-6 receptor, and this enhancement of STAT3 phosphorylation was suppressed by anti-IL-6 antibody treatment (Supplementary Fig. 5d). Consistently, hepatocyte-specific STAT3 knockout mice reportedly show retardation of liver regeneration after PHx[36]. Considering that STAT3 binds to the Foxm1 gene promoter and that a STAT3 inhibitor was shown to reduce Foxm1 expression in a leukemic cell line[27], STAT3 phosphorylated by IL-6 signaling may transactivate Foxm1 gene expression in regenerating hepatocytes.

Collectively, muscarinic signals from the vagal nerve may stimulate IL-6 production by hepatic macrophages, thereby activating the hepatocyte FoxM1 pathway in a paracrine manner. Thus, acute liver regenerative responses are achieved by a multistep mechanism comprised of neuronal, immune, and parenchymal cells. We recently reported that vagal signals directly activate the FoxM1 pathway in pancreatic β-cells, thereby enhancing compensatory proliferation[10,11]. Pancreatic islets are richly innervated by vagal nerves[11–13], and this anatomical feature may allow vagal signals to reach islet cells efficiently. In contrast, it was reported that vagal nerves in the liver were observed only around the portal region[14]. Therefore, to achieve prompt regeneration of remnant hepatocytes, there must be a distinct mechanism whereby neuronal signals are conveyed to individual cells in the liver, which is a large organ. In this context, taking advantage of resident macrophages as intermediaries, regenerative signals from the vagal nerve can be amplified and spread throughout the entire remnant liver by IL-6 secretion (Fig. 8).

The findings obtained in the present study illustrate the novel concept that a vagal signal–macrophage link promptly promotes organ regeneration which is necessary especially after severe organ injury. This concept may lead to better understanding of both the organ regenerative process and the whole-body homeostasis-maintaining mechanisms. Organ regeneration is reportedly regulated by neuronal signals in other tissues as well, such as the limbs[37,38] and pancreatic β-cells[10,11]. Macrophages also play regulatory roles in the regeneration of several tissues other than the liver, such as the spinal cord and the heart[23]. Thus, our results raise the possibility of elucidating whether the vagal signal–macrophage link is involved in the regenerations of other organs/tissues and may be useful for developing novel strategies for regenerative medicine.

## Methods

**Animals.** Animal studies were conducted in accordance with the Tohoku University institutional guidelines. Ethical approval has been obtained from the Institutional Animal Care and Use Committee of the Tohoku University Environmental & Safety Committee.

Male C57BL/6N mice were purchased from SLC Japan (Shizuoka, Japan). Transgenic mice expressing a Cre recombinase transgene fused to mutated estrogen receptor ligand-binding domains under the control of the mouse serum albumin promoter/enhancer (SA-CreER$^{T2}$)[21] were gifts from Prof. Pierre Chambon and Dr. Daniel Metzger (Institute of Genetics and Molecular and Cellular Biology, Illkirch-Cedex, France). To obtain tamoxifen-inducible liver-specific FoxM1 knockout (iFoxM1LKO) mice, we crossed SA-CreER$^{T2}$ mice and FoxM1$^{flox/flox}$ mice[22]. These mice have mixed backgrounds. At 8 weeks of age, SA-CreER$^{T2}$; FoxM1$^{flox/flox}$ mice and FoxM1$^{flox/flox}$ mice (as controls) were injected intraperitoneally with 1 mg of tamoxifen (Sigma, St. Louis, MO, USA) dissolved in corn oil (Sigma) every other day for 5 days, and were subjected to PHx 7 days after the end of tamoxifen administration. All mice were housed individually in a controlled environment (room temperature 25 °C) with a 12-h light–dark cycle, and received standard chow and drinking water ad libitum. Animal studies were conducted in accordance with Tohoku University institutional guidelines.

**Surgical procedures.** All operations were carried out on 8–10-week-old male mice, anesthetized with an intraperitoneal injection containing a mixture of medetomidine (0.3 mg/kg), midazolam (4 mg/kg), and butorphanol tartrate (5 mg/kg). For HV, after incising the abdominal wall, the stomach was retracted to expose the anterior vagal trunk and hepatic branch, and then only the hepatic branch was transected with fine forceps. Immediately after HV, 70% PHx was performed. The left lateral and median lobes of the liver were securely ligated with 6-0 silk suture and resected[39,40]. For the sham operations, only the abdominal incision was made, and the liver tissues and hepatic vagal branch were left intact. At the completion of surgery, the abdominal muscles and skin were both sutured layer by layer with 6-0 silk. In the experiments shown in Supplementary Fig. 1d, PHx was performed 10 days after HV or sham operation for HV.

**Recombinant adenovirus.** Recombinant adenoviruses carrying the human Foxm1 gene (Ad-hFoxM1) and β-galactosidase gene (Ad-LacZ) were prepared[41]. Then, $1 \times 10^8$ plaque-forming units (PFU) per mouse of the Ad-hFoxM1 were injected intravenously 1 day prior to PHx. The control mice were given $1 \times 10^8$ PFU per mouse of the Ad-LacZ. In the experiments shown in l Fig. 5d, e, recombinant adenoviruses were injected into mice immediately after control or clodronate liposome administration.

**In vivo carbachol treatment.** Carbachol (Nacalai Tesque, Kyoto, Japan) was dissolved in saline to 2 mmol/l, and mice were intraperitoneally administered 200 nmol carbachol (0.1 ml of 2 mmol/l solution). Livers were harvested 6 h after carbachol administration.

**In vivo atropine treatment.** Atropine (Sigma) was dissolved in saline with ethanol (10% v/v) to 625 mg/dl. Mice were intraperitoneally administered atropine, at 25 μg/g body weight, twice a day from 2 days before the surgical interventions until kill.

**Studies with Hepa1–6 cells.** Murine hepatocyte cell line Hepa1–6 cells (CRL-1830) were obtained from ATCC and maintained in Dulbecco's modified Eagle medium (DMEM) with 10% fetal bovine serum (FBS), 100 U/ml penicillin, and 100 μg/ml streptomycin (P/S) at 37 °C with 5% $CO_2$ and 95% air. Hepa1–6 cells were incubated with 100 μM carbachol (Nacalai Tesque) for 24 h, and then collected for RNA extraction. Water was used as the vehicle.

**Studies with peritoneal macrophages.** Peritoneal macrophages were harvested from lavage of 8-week-old male C57BL/6 N mice 4 days after intraperitoneal injection with 4% thioglycollate (Sigma)[42], plated onto 24-well plates with $1.0 \times 10^5$ cells per well and cultured in DMEM with 10% FBS and P/S. At 24 h after isolation, the cells were incubated with 100 μM carbachol (Nacalai Tesque)[43] and 100 μM atropine (Sigma) for 4 h, and finally collected for RNA extraction. The carbachol concentration was determined by examining dose-dependent effects of carbachol on IL-6 expression in primary macrophages. The atropine concentration was determined according to previously reported in vitro experiments[11].

**Macrophage depletion.** To deplete hepatic macrophages, mice were intraperitoneally administered 140 μg/body clodronate or control liposomes (FormuMax Scientific, Sunnyvale, CA, USA) 24 h prior to surgical interventions.

**Studies with primary hepatocytes**. Primary hepatocytes were isolated by a collagenase perfusion method[44] from 10-week-old iFoxM1LKO and control male mice. After making an abdominal incision, the portal vein was cannulated and the inferior vena cava was cut, the liver was then perfused with Liver Perfusion medium (Thermo Fisher Scientific, Waltham, MA, USA) at 5 ml/min for 9.5 min, followed by perfusion with HEPES buffer containing 0.5 mg/ml collagenase (Sigma) at 5 ml/min for an additional 9.5 min. The liver was excised and dissociated in DMEM with 10% FBS and P/S, filtered through a 70-μm nylon mesh, and centrifuged at 50g for 5 min three times to clear the non-parenchymal fraction. Then, fractions containing hepatocytes were suspended with 35% Percoll (GE Healthcare, UK) solution and centrifuged at 60g for 10 min. The pellets were washed and suspended with DMEM and plated onto 96-well plates with $2.5 \times 10^4$ cells per well. At 48 h after isolation, hepatocytes were incubated with 100 ng/ml recombinant murine IL-6 (PEPROTECH, London, UK) for 6 h, and then collected for RNA extraction. In experiments employing ex vivo carbachol treatment, to avoid macrophage contamination, hepatocytes were incubated with 1 mM clodronate liposomes for 1 h starting at 24 h after isolation. At 24 h after starting this treatment with clodronate liposomes, the hepatocytes were stimulated with 100 μM carbachol for 4 h. In STAT3 inhibition experiments, at 48 h after isolation, hepatocytes were incubated with 100 μM cell-permeable STAT3 inhibitor peptide (EMD Millipore, Billerica, MA, USA) for 1 h, and then stimulated with recombinant murine IL-6 for 6 h.

**RNA extraction and quantitative real-time PCR**. Total RNA was extracted from isolated macrophages, hepatocytes, and Hepa1–6 cells using an RNeasy Micro Kit (Qiagen, Hilden, Germany), and from the liver using an RNeasy Mini Kit (Qiagen). cDNA synthesis was performed with a QuantiTect Reverse Transcription Kit (Qiagen) using 100 ng of RNA from cultured cells and 1 μg of total RNA from the liver. Real-time PCR was performed using the Light Cycler Quick System 350S (Roche Diagnostic, Mannheim, Germany)[45]. The relative amount of mRNA was calculated with β-actin mRNA serving as the invariant control. The oligonucleotide primers are presented in Supplementary Table 1.

**Histological analysis**. Excised liver specimens were fixed in 10% formalin, embedded in paraffin, and sectioned. Sections were stained with hematoxylin and eosin. For bromodeoxyuridine (BrdU) in situ detection, mice were injected intraperitoneally with 1 mg of BrdU (BD Bioscience, San Jose, CA, USA) diluted to 10 mg/ml with phosphate-buffered saline 2 h before liver extraction. The labeled cells were immunostained with anti-BrdU antibody (51-75512 L, BD Bioscience). The number of BrdU-positive nuclei per 1000 hepatocytes per liver specimen was counted. For detection of macrophages, liver sections were stained with anti-F4/80 antibody (12-4801-80, Affymetrix eBioscience, Santa Clara, CA, USA).

**Immunoblotting**. Liver samples were homogenized in lysis buffer containing 100 mM Tris, pH 8.5, 250 mM NaCl, 1 mM EDTA, 1 mM phenylmethylsulfonyl fluoride, aprotinin at 1:5000 dilution, and leupeptin at 1:5000 dilution[46]. Tissue homogenates were centrifuged and the supernatants including tissue protein extracts were boiled in Laemmli buffer containing 10 mM dithiothreitol, then subjected to SDS-polyacrylamide gel electrophoresis. Separated proteins were transferred to nitrocellulose membranes and blocked in Tris-buffered saline containing 3% FBS. Immunoblot analyses were performed using antibodies to phospho-STAT3 (Tyr705) (#9131, Cell Signaling Technology, Danvers, MA, USA), total STAT3 (#4904, Cell Signaling Technology), phospho-STAT1 (Tyr701) (#9167, Cell Signaling Technology), phospho-STAT5 (Tyr694) (#9314, Cell Signaling Technology), FoxM1 (13147-1-AP, Proteintech, Rosemont, IL, USA), Cyclin A2 (ab181591, Abcam, Cambridge, UK), Cdk1 (#28439, Cell Signaling Technology), and PLK1 (#4535, Cell Signaling Technology) at 1:2000 dilution and actin (A2066, Sigma) at 1:5000 dilution. Quantitative data were obtained employing a ChemiDoc Touch Imaging System (Bio-Rad Laboratories, Hercules, CA, USA). Uncropped membrane images are presented in Supplementary Fig. 6.

**Plasma IL-6 measurement**. Blood samples were collected from tail veins. Plasma IL-6 concentrations were measured using a Mouse IL-6 ELISA kit (R&D Systems, Minneapolis, MN, USA) according to the manufacturer's instructions.

**In vivo IL-6 neutralization**. We administered 500 μg/body neutralizing antibodies to IL-6 and control rat IgG (Bio X Cells, West Lebanon, NH, USA) intraperitoneally 1 hr before PHx.

**Statistical analysis**. All data are presented as means ± standard error. For experiments in which data differences among three or four groups needed to be assessed, we used one-way ANOVA followed by Bonferroni's post hoc test. In experiments in which data differences between two groups were assessed, results were analyzed using the unpaired t test. Survival curves were estimated using the Kaplan–Meier method and differences in survival among the groups were tested by applying the log-rank test with Bonferroni correction. Differences were considered to be significant at $p < 0.05$. No statistical method was used to predetermine sample size. Most sample sizes were chosen based on data from previous publications. Statistical analyses were performed assuming a normal distribution in all experiments.

## Data availability

The data that support the findings of this study are available from the corresponding author upon reasonable request. A Reporting Summary for this Article is available as a Supplementary Information file.

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

## Acknowledgements

Prof. Pradip Raychaudhuri of the University of Illinois at Chicago contributed to generation of the FoxM1-floxed mice. Prof. Pierre Chambon and Prof. Daniel Metzger of the Institute of Genetics and Molecular and Cellular Biology, Illkirch-Cedex contributed to generation of the SA-CreER$^{T2}$ mice. This work was supported by Grants-in-Aid for Scientific Research to H.K. (17H01565 and 16K15486), J.I. (17K09816), and T.I. (18K16221) from the Japan Society for the Promotion of Science, the CREST to H.K. (17gm0610001h0006) from the Ministry of Education, Culture, Sports, Science, and Technology of Japan, Takeda Science Foundation to J.I. This research was also supported by the Japan Agency for Medical Research and Development, AMED, under Grant Numbers JP17gm0610001 and JP17gm5010002 to H.K. and the PRIME to J.I. (18gm6210002h0001). We thank T. Takasugi, J. Fushimi, H. Hatakeyama, and A. Iwama (all belong to the Department of Metabolism and Diabetes, Tohoku University Graduate School of Medicine) for technical support.

## Author contributions

T.I. and J.I. conducted the research and obtained the data, contributed to relevant discussions, wrote the manuscript, and reviewed/edited the paper. J.Y., Y.K., A.E., H.S., M.K., Y.A., K.T., S.K., K.K., J.G., K.U., S.S., Y.I. and T.Y. contributed to the relevant discussions. V.V.K. provided FoxM1-floxed mice. H.K. contributed to the relevant discussions, writing the paper, and reviewing/editing the paper.

## Additional information

**Competing interests:** The authors declare no competing interests.

