## [Peer Review File · Nature Communications]

Reviewers' comments:

Reviewer #1 - Liver regeneration and IL-6 (Remarks to the Author):

The authors present interesting results on the effects of hepatic vagotomy (HV) on liver regeneration. The effect is connected to regulation of expression of FoxM1 and evidence is presented that this effect is mediated by macrophages. The data are thoroughly analyzed and well presented. There are, however, several issues that need to be addressed, as follows:

1. Fig. 1A demonstrates that there is increased mortality in PHx-HV compared to simple PHx, and that animals that survive beyond Day 3 have regular survival thereafter. The reasons for the enhanced mortality are not explained and it is implied that this is due to a failed regeneration. Histology of the liver of the affected animals is not provided, and it should (in Supplement). It is quite possible that data Fig. 1A can be explained not as a result of HV effects on regeneration, but simply as a result of operation-related non-specific injury. The authors should also present data on mortality of normal, non-hepatectomized, mice, subjected to HV.

2. There is recent literature documenting that in addition to macrophages, hepatocytes also express IL6 (Norris CA et al., PLoS1, 2014, vol 9, issue 4, PMID: 24763697. The authors have not addressed the possibility that some of the effects assumed to be mediated by macrophages may actually be mediated by hepatocytes. Primary cultures of hepatocytes should be tested in that regard. In fact, it is possible that the high levels of FoxM1 shown in the hepatocytes cultures may be due to IL6 secretion by hepatocytes.

3. The manuscript shows that administration of FoxM1 (using adenoviral vectors) in mice subjected to PHx-HV reverses the mortality seen with PHx-HV. The data are true, but also potentially non-related. Administration of FoxM1 will have its own effects, unrelated to HV, and those effects may supersede the effect of HV. The authors should administer FoxM1 in normal mice, not subjected to PHx or HV, and assess the effects of that in terms of hepatocyte proliferation. The same should also be done in normal mice with administering carbachol, or some tolerated cholinergic agent. Would that stimulate expression of FoxM1 in hepatocytes or production of IL6 in hepatic macrophages?

4. Data shown in Fig. 1 and Supplemental Fig. 1 A demonstrate practically equal liver weights at Day 7 between PHx and PHx-HV. Data in both figures should be expanded to also demonstrate the situations in intermediate (3, 5) days. It is important to know how fast the recovery of liver weight occurs in PHx-HV, especially since all evidence of enhanced mortality lasts only until day 2 after PHx-HV.

Reviewer #2 - FoxM1 (Remarks to the Author):

In this paper, Izumi et al. have discovered a vagus-macrophage-hepatocyte link in the regulation of acute liver regeneration immediately after liver injury and has suggested that this system is essential for survival.

Briefly, the authors have shown that hepatic branch vagotomized mice has lower acute survival rate within 3 days post-operation as compared to PHx-sham mice. Hepatic branch vagotomy inhibited acute hepatocyte proliferation and consequently increased mortality, as suggested by the significantly lower % of BrdU-positive hepatocytes in vagotomized mice group as compared to the sham group on the second day post-surgery. It is also shown that this vagus-controlled acute liver injury response is FoxM1-dependent, as the relative expression levels of FoxM1, FoxM1 downstream target genes and MKi67 were significantly hampered in vagotomized mice.

Additionally, acute liver regenerative responses after PHx were compromised in liver-specific FoxM1 knockout mice, whereas hepatic FoxM1 supplementation in vagotomized mice restored liver regeneration and significantly reconstructed cumulative survival.

Furthermore, it is also indicated that resident macrophages mediate this vagus-hepatocyte link via IL-6 production in a paracrine fashion. Hepatic macrophage deletion suppressed post-PHx Foxm1 upregulation and cellular proliferation in liver sections and increased mortality. Hepatic IL-6 was upregulated significantly after PHx-sham and this increase was inhibited by hepatic branch vagotomy, muscarinic blockade or resident macrophage depletion. Lastly, IL-6 inhibition prevented Foxm1 upregulation and cellular proliferation in the liver section.

Overall, the data presented are relatively novel and compact. However, the role of FOXM1 in liver regeneration has previously been established. There are a few suggestions that the authors can also consider to make the work more substantial and novel. At the moment, the study presented is more correlational and lacks novel mechanistic findings. The signalling pathways and components involved are not explored and validated.

It is as yet elusive the exact mechanism where vagal signals are essential for post-PHx acute survival in mice. The reason as to why vagal signals are critical for acute liver regeneration can be explored deeper. The authors have not shown the exact mechanism by which FOXM1 is regulated by IL-6. It would not be such a novel finding if it is mediated through the receptor-PI3K-Akt-FOXO signalling pathway.

The interleukin (IL)-6 family cytokines are a group of cytokines consisting of IL-6, IL -11, ciliary neurotrophic factor (CNTF), leukemia inhibitory factor (LIF), oncostatin M (OSM), cardiotrophin 1 (CT-1), cardiotrophin-like cytokine (CLC), and IL-27, mainly released by T cells and macrophages as pro-inflammatory factors. This family of inflammatory cytokines have overlapping but also distinct biologic activities and are involved among others in the regulation of the hepatic acute phase reaction, in B-cell stimulation, in the regulation of the balance between regulatory and effector T cells, in metabolic regulation, and in many neural functions. There is a possibility that other IL-6 factors are involved in this regulation, therefore their expression levels, including the sole receptors, gp130, should be assessed. Lastly, IL-6 family downstream effectors include STAT1, STAT3 and STAT5, also this activation of this signalling pathway is highly dependent on homodimer or heterodimer formation of the STAT molecules, hence the activity/expression of all should be taken into consideration.

All the experiments were attributed to mice studies, it is worth checking the relevance of this vagus-macrophage-hepatocyte link in human cells, patient samples or clinical data in order to discover the possibility of such mechanism in human acute liver regeneration.

More specific points:

Figure 1C Day 7 graph, the difference between SO and PHx-sham BrdU-positive hepatocytes (%) does not seem to be that big. Nonetheless, the author said that it was of significant difference. Any difference was not clearly shown in the figure, perhaps due to the scale of the graph? Suggestion: for the Day 7 graph, scale bar range should be lower.

Figure 4A and B. Figure 4A showed both PHx-sham and PHx-HV data, but 4B only showed control (PHx) and Clodronate (PHx treated with Clodronate). Why did the author not do the PHx-HV + Clodronate? It would be interesting to see whether PHx-HV + Clodronate have more, less, or on effect on IL-6 expression than the PHx + Clodronate alone to determine the role of HV

Figure 4C and 4D. Ex vivo experiment of primary macrophages treated with carbachol and carbachol+atropine, but in vivo experiment, only atropine alone was used. Why?

Figure E the relative expression of FOXM1, Ccna2, and Cdk1 seemed to increase in control that were treated with IL-6, however, the increases seem rather small to be of very significant as the author suggested.

Figure 4F and Supplemental Figure 5B, there was no tubulin/control for western blot.

Reviewer #3 - neuro-immune communication (vagus nerve)(Remarks to the Author):

In this manuscript Izumi et al demonstrate a role for the vagus nerve in promoting the initial stages of hepatic regeneration and survival in a murine model of partial hepatectomy, and delineate cellular and molecular mechanisms involved. They show that hepatic Foxm1 levels are significantly elevated in mice with hepatectomy and hepatic vagotomy suppresses this increase and increases mortality. They also substantiate their findings about the role of Foxm1 in this regulation by utilizing inducible liver specific FoxM1 knockout mice. They further show a role for resident macrophages and the release of IL-6 as a link between vagal cholinergic signaling and hepatocyte proliferation by utilizing macrophage deletion and pharmacological cholinergic modalities (e.g. an antagonist of muscarinic cholinergic receptors-atropine, and an agonist-carbacol). These findings provide an important mechanistic insight into the role of vagus nerve cholinergic signaling in facilitating liver regeneration and improving survival following severe liver injury.

Main concerns/issues:

1. The rationale for doing hepatic vagotomy (i.e. the surgical transection of the hepatic branch of the vagus nerve) was apparently to interrupt/eliminate vagus nerve signaling to the liver. However, the hepatic vagotomy was performed immediately before hepatectomy. It is known that surgical transection causes powerful mechanical stimulation of the nerve with duration and physiological consequences which may be long lasting. Therefore, to avoid this acute stimulatory effect caused by the procedure, in many previous studies, especially in the context of studying the anti-inflammatory role of the vagus nerve, unilateral cervical or bilateral subdiaphragmatic vagotomy (PMID: 28065837; PMID: 25063706; PMID: 16785311) was performed at least a few days prior to including these animals in experiments. How did the authors account for the real possibility of stimulating hepatic vagus nerve activity by performing hepatic vagotomy in their experiments?
2. The authors emphasize the role of IL-6 and study IL-6 as a major mediator produced by activated macrophages in the liver in the link between vagus nerve cholinergic output and hepatocyte regeneration. Were results from hepatectomy and hepatic vagotomy experiments performed in IL-6 knockout mice previously described in the literature?
3. Summarizing some of their results the authors conclude that "Collectively, vagus-derived cholinergic signals can directly upregulate macrophage IL-6 production through a muscarinic receptor-dependent mechanism." Experiments with carbacol and atropine were performed with just one concentration of each of the compounds. How was this concentration chosen and did the authors examine/observe dose-dependent effects? If we put things in a broader context, one of the seminal papers demonstrating the role of the vagus nerve in controlling inflammation the authors refer to (PMID: 10839541), actually shows that vagus nerve stimulation suppresses hepatic TNF levels in endotoxemic rats. There are other papers demonstrating the efficacy of cholinergic modalities in suppressing IL-6 release too. The authors indeed refer to some work on the anti-inflammatory role of cholinergic signaling mediated through the alpha 7 nicotinic acetylcholine receptor and demonstrate that the increase in IL-6 levels in their experiments is mediated through muscarinic receptors. However, in vivo, vagus nerve cholinergic signaling will hit both muscarinic and nicotinic receptors expressed on macrophages. In addition, there is experimental evidence that centrally-acting muscarinic receptor agonists have powerful anti-inflammatory properties in murine endotoxemia (PMID: 16549778; PMID: 25063706), IBD (PMID: 25295619) and other conditions and these central effects are linked to the efferent vagus

nerve-based cholinergic anti-inflammatory pathway. Are the authors suggesting that cholinergic signaling through brain and peripheral muscarinic receptors have different effects on cytokine readouts? The authors should comment on their findings in this broader context and provide plausible explanations.

4. Relevant to the points above: Is there a systemic inflammatory response in the context of hepatectomy and did the authors measure systemic levels of IL-6, TNF and other cytokines and the effect of vagotomy?

5. I also have some concerns about the low n-number used (4 and even 3 in some experiments). Were any power calculations used to decide the n numbers? The statistical tests that were used assume a normal distribution of the data, which is often not the case when very small n-numbers (such as n=4) are used. How did the authors test for normality of the data? If not, they should mention that and note that statistical analysis was performed assuming normal distribution.

Minor concerns:

The discussion is written in a way that sounds somewhat repetitive (pointing to Figures, which is not the typical way) with what is already said in the result section. I'd suggest reorganizing it to avoid the overlap with "Results".

Responses to reviewer 1

1. Fig. 1A demonstrates that there is increased mortality in PHx-HV compared to simple PHx, and that animals that survive beyond Day 3 have regular survival thereafter. The reasons for the enhanced mortality are not explained and it is implied that this is due to a failed regeneration. Histology of the liver of the affected animals is not provided, and it should (in Supplement). It is quite possible that data Fig. 1A can be explained not as a result of HV effects on regeneration, but simply as a result of operation-related non-specific injury. The authors should also present data on mortality of normal, non-hepatectomized, mice, subjected to HV.

As the reviewer suggested, we histologically analyzed the livers of PHx-HV-mice immediately after their deaths. In contrast to PHx-sham-mice, focally necrotic areas with marked bleeding were diffusely observed in the livers of these mice, consistent with findings of severe liver failure. Therefore, vagal signal-mediated prompt regenerative responses are likely required for prevention of liver failure development after PHx. These findings are now presented in Supplementary Figure S1A and are described in the Results section of the revised manuscript (page 5, lines 7 to 10).

In accordance with the reviewer's request, we added the mortality data of the non-hepatectomized mice, subjected only to HV. Post-operative death after HV alone was very rare, ruling out the possibility that high post-operative mortality in PHx-HV-mice is due to the HV procedure. These findings are now presented as HV-alone in Figure 1A and described in the Results section of the revised manuscript (page 5, lines 6 to 7).

2. There is recent literature documenting that in addition to macrophages, hepatocytes also express IL6 (Norris CA et al., PLoSI, 2014, vol 9, issue 4, PMID: 24763697. The authors have not addressed the possibility that some of the effects assumed to be mediated by macrophages may actually be mediated by hepatocytes. Primary cultures of hepatocytes should be tested in that regard. In fact, it is possible that the high levels of FoxM1 shown in the hepatocytes cultures may be due to IL6 secretion by hepatocytes.

To test the effects of cholinergic signals on hepatocyte IL-6 production, we treated primary hepatocytes with carbachol. To eliminate macrophage contamination of primary hepatocytes, we pre-treated cells, primarily cultured from the liver, with clodronate. Under this condition, carbachol treatment did not increase *Il-6* expression. Therefore, it is unlikely that cholinergic signals enhance IL-6 production from hepatocytes. These

findings are now presented in Supplementary Figure S4D and are described in the Results section of the revised manuscript (page 11, lines 1 to 2). We also added the method of macrophage depletion of primary hepatocytes to the Methods section (page 21, lines 7 to 11). Importantly, as shown in Figure 4B of the original and revised manuscripts, macrophage depletion *in vivo* almost completely inhibited the increase in hepatic *Il-6* expression 6 hours after PHx, clearly indicating the source of IL-6 after PHx to be macrophages. Taken together, these observations indicate that the involvement of hepatocyte-derived IL-6 in vagal signal-mediated acute liver regeneration is unlikely.

3. The manuscript shows that administration of FoxM1 (using adenoviral vectors) in mice subjected to PHx-HV reverses the mortality seen with PHx-HV. The data are true, but also potentially non-related. Administration of FoxM1 will have its own effects, unrelated to HV, and those effects may supersede the effect of HV. The authors should administer FoxM1 in normal mice, not subjected to PHx or HV, and assess the effects of that in terms of hepatocyte proliferation. The same should also be done in normal mice with administering carbachol, or some tolerated cholinergic agent. Would that stimulate expression of FoxM1 in hepatocytes or production of IL6 in hepatic macrophages?

We thank the reviewer for this insightful comment. We administered FoxM1 adenovirus to normal mice, not subject to either PHx or HV. In contrast to the case of PHx+HV mice, no increases in expressions of endogenous mouse *FoxM1* and its target genes as well as *Mki67* in the liver were observed in normal mice. In addition, carbachol administration to normal mice, not subject to either PHx or HV, did not increase hepatic expressions of these FoxM1-related genes. These results indicate that the vagus-macrophage-hepatocyte linking mechanism alone does not elicit proliferative responses of hepatocytes under normal conditions, and suggest that additional and unknown signals, induced by PHx, are necessary for promoting hepatocyte proliferation. The linking mechanism, which we herein elucidated, functions especially under the situation wherein prompt hepatocyte regenerative responses are required, such as after PHx. These findings are now presented in Supplemental Figures S3B and S3C, and described in the Results section of the revised manuscript (page 8, lines 11 to 15), as well as being discussed in the Discussion section of the revised manuscript (page 13, lines 9 to 15). We also added the method of *in vivo* carbachol treatment to the Methods section (page 19, lines 13 to 16).

4. Data shown in Fig. 1 and Supplemental Fig. 1 A demonstrate practically equal liver weights at Day 7 between PHx and PHx-HV. Data in both figures should be expanded to also demonstrate the situations in intermediate (3, 5) days. It is important to know how fast the recovery of liver weight occurs in PHx-HV, especially since all evidence of enhanced mortality lasts only until day 2 after PHx-HV.

As suggested by the reviewer, we examined recovery of hepatic weights, gene expressions and hepatocyte proliferation on day 5 after PHx. The weights of livers from both PHx-sham- and PHx-HV-mice were remarkably increased on day 5, with the organ being heavier than those on day 7, as well as appearing their edematous. The livers of PHx-HV-mice tended to weigh slightly less than those of PHx-sham-mice but the differences did not reach statistical significance. In contrast, on day 7, the liver weights were quite similar in PHx-sham- and PHx-HV-mice. Consistent with hepatic weight changes after PHx, the difference in BrdU-positive hepatocyte ratios between PHx-sham- and PHx-HV-mice lost significance on days 5 and 7. Thus, the vagal signal-independent mechanism of liver recovery may become dominant after post-operative day 3. These results further strengthen our conclusion that vagal signals selectively play an important role in acute-phase liver regeneration, which is critical for post-PHx survival. These findings were added to Figures 1B and 1C and supplemental Figures S1B and S1G and are also described in the Results section of the revised manuscript (page 5, lines 14 to 15, page 5, line 23 to page 6, line1 and page 7, lines 4 to 6).

Responses to reviewer 2

It is as yet elusive the exact mechanism where vagal signals are essential for post-PHx acute survival in mice. The reason as to why vagal signals are critical for acute liver regeneration can be explored deeper. The authors have not shown the exact mechanism by which FOXM1 is regulated by IL-6. It would not be such a novel finding if it is mediated through the receptor-PI3K-Akt-FOXO signalling pathway.

The interleukin (IL)-6 family cytokines are a group of cytokines consisting of IL-6, IL-11, ciliary neurotrophic factor (CNTF), leukemia inhibitory factor (LIF), oncostatin M (OSM), cardiotrophin 1 (CT-1), cardiotrophin-like cytokine (CLC), and IL-27, mainly released by T cells and macrophages as pro-inflammatory factors. This family of inflammatory cytokines have overlapping but also distinct biologic activities and are involved among others in the regulation of the hepatic acute phase reaction, in B-cell stimulation, in the regulation of the balance between regulatory and effector T cells, in

metabolic regulation, and in many neural functions. There is a possibility that other IL-6 factors are involved in this regulation, therefore their expression levels, including the sole receptors, gp130, should be assessed. Lastly, IL-6 family downstream effectors include STAT1, STAT3 and STAT5, also this activation of this signalling pathway is highly dependent on homodimer or heterodimerformation of the STAT molecules, hence the activity/expression of all should be taken into consideration.

To explore the molecular mechanism by which IL-6 activates the hepatocyte FoxM1 pathway in more detail, we pretreated primary hepatocytes with STAT3 inhibitor peptide, which inhibits both homodimerization of STAT3 and heterodimerization of STAT3 and STAT1, followed by treatment with IL-6. Pretreatment with the STAT3 inhibitor almost completely blocked IL-6-mediated increases in *FoxM1* and its target genes as well as *Mki67*. Thus, the STAT3 pathway, rather than PI3K-Akt-FOXO signaling, is involved in the FOXM1 regulation by IL-6. In addition, we examined the phosphorylation of STAT1 as well as that of STAT5 after PHx. In contrast to STAT3, phosphorylations of STAT1 and STAT5 showed no alterations after PHx. These results strongly suggest that IL-6 enhances activation of STAT3, rather than that of STAT1 or STAT5, thereby up-regulating FoxM1-related genes in hepatocytes. These findings are now presented in supplemental Figures S5B and S5C and described in the Results section of the revised manuscript (page 12, lines 1 to 9).

In addition, as the reviewer suggested, we examined expressions of IL-6 family cytokines as well as *gp130* in the livers of PHx-sham and PHx-HV mice 6 hours after surgery. As shown in the figure below attached to our responses, however, no cytokines yielded results indicating that PHx up-regulates their expressions, while HV suppresses the PHx-induced up-regulations. In particular, hepatic expression of *gp130* was not altered between PHx-sham and PHx-HV mice. Therefore, the involvement of these cytokines in the vagus-macrophage-hepatocyte link is unlikely.

Figure. Relative gene expression levels of IL-6 family cytokines and gp130

8-week-old male C57BL/6N mice were subjected to sham operation for PHx (SO) (n = 6), PHx concomitantly with sham operation for HV (PHx-sham) (n = 6), or PHx with HV (PHx-HV) (n = 6). Liver specimens were collected 2 days after the operation, followed by extraction of total RNA. cDNA was then synthesized by reverse transcription using 1 µg RNA, and real-time PCR was performed to determine relative gene expression levels of IL-6 family cytokines and gp130. The relative amounts of mRNA were calculated with β-actin mRNA serving as the invariant control.

*P < 0.05; **P < 0.01 assessed by one-way ANOVA followed by Bonferroni's post hoc test. n.s., not significant.

All the experiments were attributed to mice studies, it is worth checking the relevance of this vagus-macrophage-hepatocyte link in human cells, patient samples or clinical data in order to discover the possibility of such mechanism in human acute liver regeneration.

We agree with the reviewer's comment indicating that it is worth examining the vagus-macrophage-hepatocyte link in human, especially patient, samples. We think, however, that to draw a conclusion regarding possible clinical relevance might be

beyond the scope of the present study. As the next step toward applying our present findings to clinical fields, we will tackle this important issue in the near future.

More specific points

Figure 1C Day 7 graph, the difference between SO and PHx-sham BrdU-positive hepatocytes (%) does not seem to be that big. Nonetheless, the author said that it was of significant difference. Any difference was not clearly shown in the figure, perhaps due to the scale of the graph? Suggestion: for the Day 7 graph, scale bar range should be lower.

We apologize for the confusing data presentation. According to the reviewer's suggestion, we added the figures in which the scale bar ranges were lowered in framed boxes of Figure 1C.

Figure 4A and B. Figure 4A showed both PHx-sham and PHx-HV data, but 4B only showed control (PHx) and Clodronate (PHx treated with Clodronate). Why did the author not do the PHx-HV + Clodronate? It would be interesting to see whether PHx-HV + Clodronate have more, less, or no effect on IL-6 expression than the PHx + Clodronate alone to determine the role of HV.

As the reviewer suggested, we examined the effects of HV on *Il-6* expression after PHx in clodronate liposome-treated mice. Consistent with the results of hepatocyte proliferative responses (Figures 3E and 3F of both the original and the revised manuscripts), HV failed to exert further inhibitory effects on upregulation of *Il-6* expressions in clodronate liposome-treated mice. These results further strengthen our conclusion that vagal signals activate the hepatic FoxM1 pathway through the macrophage-IL-6-dependent mechanism. These findings were added to Figure 4B, and described in both the Results (page 10, lines 14 to 17) and the Discussion (page 14, line 22) sections of the revised manuscript.

Figure 4C and 4D. Ex vivo experiment of primary macrophages treated with carbachol and carbachol+atropine, but in vivo experiment, only atropine alone was used. Why?

As shown in Figures 4A and 4D of both the original and the revised manuscripts, PHx induced increases in hepatic *IL-6* expression, and HV or atropine treatment significantly suppressed *IL-6* upregulations. These results clearly showed that cholinergic signals

elicited by vagal nerves were already enhanced in the remnant liver after PHx. Therefore, it is unlikely that further stimulation with carbachol in the *in vivo* experiment (Figure 4D) would provide additional information showing the significance of endogenous cholinergic signals in the liver.

Figure E the relative expression of FOXM1, Ccna2, and Cdk1 seemed to increase in control that were treated with IL-6, however, the increases seem rather small to be of very significant as the author suggested.

As the reviewer pointed out, the effects of IL-6 on the expressions of *FoxM1* and its target genes were weak in primary hepatocytes. This was due to much higher baseline *FoxM1* levels in primary hepatocytes than hepatocytes in non-treated quiescent livers. We speculate that the isolation procedure may enhance *FoxM1* expression in primary hepatocytes. We acknowledged this limitation in the original and revised manuscripts (page 11, lines 9 to 11). However, importantly, even under these conditions, IL-6 treatment further increased expressions of *FoxM1* and its target genes as well as *MKi67*, indicating substantial proliferative effects of IL-6 on hepatocytes.

Figure 4F and Supplemental Figure 5B, there was no tubulin/control for western blot.

We re-blotted the sheets with anti-actin antibody as a control. Actin expressions were similar in SO, PHx-sham and PHx-HV samples. We added the western blotting images of actin to Figure 4F and Supplemental Figure S5D of the revised manuscript. In addition, we included images of actin as a control in the new western blotting images shown in Supplemental Figure S5C of the revised manuscript. We also present the uncropped western blotting images in Supplemental Figure S7.

Responses to reviewer 3

Main concerns/issues:

1. The rationale for doing hepatic vagotomy (i.e. the surgical transection of the hepatic branch of the vagus nerve) was apparently to interrupt/eliminate vagus nerve signaling to the liver. However, the hepatic vagotomy was performed immediately before hepatectomy. It is known that surgical transection causes powerful mechanical stimulation of the nerve with duration and physiological consequences which may be long lasting. Therefore, to avoid this acute stimulatory effect caused by the procedure, in

many previous studies, especially in the context of studying the anti-inflammatory role of the vagus nerve, unilateral cervical or bilateral subdiaphragmatic vagotomy (PMID:28065837; PMID: 25063706; PMID:16785311) was performed at least a few days prior to including these animals in experiments. How did the authors account for the real possibility of stimulating hepatic vagus nerve activity by performing hepatic vagotomy in their experiments?

To explore the possibility that stimulatory effects caused by vagotomy affect hepatocyte proliferation after PHx, we performed vagotomy 10 days before PHx and examined the hepatocyte proliferation after PHx. Similar to the results obtained when hepatic vagotomy was performed immediately before hepatectomy, hepatocyte proliferation was almost completely blocked by vagotomy. Therefore, it is unlikely that the vagotomy procedure elicits unexpected effects on hepatocyte proliferation after PHx. These findings are now presented in supplemental Figure S1C and described in the Results section of the revised manuscript (page 5, lines 22 to 23).

2. The authors emphasize the role of IL-6 and study IL-6 as a major mediator produced by activated macrophages in the liver in the link between vagus nerve cholinergic output and hepatocyte regeneration. Were results from hepatectomy and hepatic vagotomy experiments performed in IL-6 knockout mice previously described in the literature?

Retardation of liver regeneration after PHx was previously reported in IL-6 knockout mice (Ref #26 of both the original and the revised manuscripts). However, nothing was known about the link between vagal nerve signals and IL-6. In our present study, antibody-mediated temporal inhibition of IL-6 after PHx *in vivo*, not congenital knockout of IL-6, markedly blunted increases in *FoxM1* and its target genes as well as *Mki67* after PHx (Figure 4G in the original and revised manuscripts). These results clearly indicate a critical role of IL-6 in activation of the FoxM1 pathway after PHx. We concluded that IL-6 is an essential mediator which transmits proliferation signals from macrophages to hepatocytes in the vagus-macrophage-hepatocyte linking mechanism.

3. Summarizing some of their results the authors conclude that “Collectively, vagus-derived cholinergic signals can directly upregulate macrophage IL-6 production through a muscarinic receptor-dependent mechanism.” Experiments with carbachol and atropine were performed with just one concentration of each of the compounds. How

was this concentration chosen and did the authors examine/observe dose-dependent effects?

We determined carbachol concentrations by examining dose-dependent effects of carbachol on IL-6 production from primary macrophages. Carbachol increased IL-6 production by primary macrophages in a dose-dependent manner, and 100 μ M of carbachol substantially enhanced IL-6 production by macrophages. This is the same concentration commonly used for cholinergic stimulation (PMID:7588223 ref #42 newly cited in the revised manuscript). These results are now presented in Supplemental Figure S4C and described in the Results section (page 10, lines 23 to 24).

We used atropine at the same concentration as that employed in our previously reported *ex vivo* experiments (Ref #11). As shown in Figure 4C of both the original and the revised manuscripts, atropine markedly blunted the increases in carbachol-induced IL-6 expression in primary macrophages. Therefore, the concentration of atropine we used is sufficient to inhibit the muscarinic effects on IL-6 production by macrophages.

How we selected the concentrations of carbachol and atropine is now described in the Methods section (page 20, lines 9 to 12).

4. If we put things in a broader context, one of the seminal papers demonstrating the role of the vagus nerve in controlling inflammation the authors refer to (PMID: 10839541), actually shows that vagus nerve stimulation suppresses hepatic TNF levels in endotoxemic rats. There are other papers demonstrating the efficacy of cholinergic modalities in suppressing IL-6 release too. The authors indeed refer to some work on the anti-inflammatory role of cholinergic signaling mediated through the alpha 7 nicotinic acetylcholine receptor and demonstrate that the increase in IL-6 levels in their experiments is mediated through muscarinic receptors. However, in vivo, vagus nerve cholinergic signaling will hit both muscarinic and nicotinic receptors expressed on macrophages. In addition, there is experimental evidence that centrally-acting muscarinic receptor agonists have powerful anti-inflammatory properties in murine endotoxemia (PMID:16549778; PMID: 25063706), IBD (PMID:25295619) and other conditions and these central effects are linked to the efferent vagus nerve-based cholinergic anti-inflammatory pathway. Are the authors suggesting that cholinergic signaling through brain and peripheral muscarinic receptors have different effects on cytokine readouts? The authors should comment on their findings in this broader context and provide plausible explanations.

We thank the reviewer for these insightful and useful comments. As the reviewer noted, there have been several reports showing activation of muscarinic signals in the central nervous system to exert systemic anti-inflammatory properties. In the reports which the reviewer pointed out, direct targets of muscarinic agonists were shown to be neuronal cells in the central nervous system, but not macrophages. In addition, the efferent vagal nerve-mediated anti-inflammatory responses in these studies were found to be independent from peripheral muscarinic effects (PMID: 16549778) but did, in fact, depend on nicotinic effects on splenic immune cells (PMID: 25295619). Thus, these previous reports showed that nicotinic, rather than muscarinic, signals have anti-inflammatory effects on peripheral macrophages. Meanwhile, since atropine treatment blunted the carbachol-mediated and PHx-induced upregulations of *IL-6* in macrophages *ex vivo* and in the remnant liver *in vivo*, respectively, muscarinic signaling is involved in liver regeneration under our study conditions. We discussed this important issue (page 15, lines 4 to 7) and added new references (Ref# 33 and 34) to the revised manuscript.

We agree with the reviewer's suggestion that cholinergic signaling would affect both muscarinic and nicotinic receptors on macrophages. In this context, we speculate that expression levels of muscarinic and nicotinic receptors on macrophages could be altered according to physiological or pathological situations, although further examinations are needed to elucidate the molecular mechanism underlying opposite responses of macrophages to cholinergic stimulation. We added these discussions (page 15, lines 14 to 17) to the revised manuscript.

5. Relevant to the points above: Is there a systemic inflammatory response in the context of hepatectomy and did the authors measure systemic levels of IL-6, TNF and other cytokines and the effect of vagotomy?

We examined the time courses of plasma concentrations of IL-6 in PHx-sham- and PHx-HV-mice. Plasma IL-6 concentrations after PHx were increased and these increases were suppressed by vagotomy, consistently with the expressions of hepatic *IL-6* in PHx-sham- and PHx-HV-mice. These results are now presented in Supplemental Figure S4B and described in the Results section (page 10, lines 11 to 13). We also added the method of plasma IL-6 measurement to the Methods section (page 22, line 23 to page 23, line 3).

6. I also have some concerns about the low n-number used (4 and even 3 in some experiments). Were any power calculations used to decide the n numbers? The statistical tests that were used assume a normal distribution of the data, which is often not the case when very small n-numbers (such as n=4) are used. How did the authors test for normality of the data? If not, they should mention that and note that statistical analysis was performed assuming normal distribution.

We performed further experiments for Figures 1C, 4B and Supplemental Figure S1F and added the higher n-numbers. However, n-numbers are still 4 in several experiments described in the revised manuscript and we did not perform power calculations. Therefore, we stated that statistical analyses were performed assuming a normal distribution in all experiments in the Methods section (page 23, lines 16 to 17).

Minor concerns:

1. The discussion is written in a way that sounds somewhat repetitive (pointing to Figures, which is not the typical way) with what is already said in the result section. I'd suggest reorganizing it to avoid the overlap with "Results".

According to the reviewer's suggestion, we removed several repetitive portions of the text, which referred to Figures, from the Discussion section of the original manuscript.

Reviewers' comments:

Reviewer #1 (Remarks to the Author):

The histology of focal necrosis after PHx and HV is most consistent with "biliary infarcts", seen after rupture of portal ductules. The authors should mention that as a also a distinct possibility that needs at some future point to be explained.

Reviewer #2 (Remarks to the Author):

The manuscript has been improved by the revision. The authors have altered the figures and did new experiments to assess the cytokines family to address some of my previous concerns. Although it is of good rationale and supportive experiment results and all of which seem to be conclusive, the study is still more correlational and lacks novel mechanistic findings. There are also some minor missing information. Additional experiments for Ad-hFoxM1 should be performed in many areas.

Results –

- Most of the results regarding FoxM1 and its target genes concentrate in the mRNA expression level in liver. FoxM1, as a transcription factor, has post-transcriptional modification and thus, mRNA level does not give an actual representation of activity. As such, its target genes may not be the direct effects of increased FoxM1 activity in liver. The reviewer would recommend some western blots on the FoxM1 and its target genes. On top of the use of tamoxifen-inducible liver-specific FoxM1 knock out mice (iFoxM1LKO), using FoxM1 selective inhibitor might be more specific as an administration of tamoxifen could potentially induce nonspecific noises that affect the results. With this in concerned, adenoviral FoxM1 supplement + FoxM1 inhibitor could be used to confirm the FoxM1 activity.

Vagal signals are critical for post-PHx survival

- It is mentioned that "Until day 5, the weights of livers from both PHx-sham- and PHx-HV-mice were remarkably increased, with an edematous appearance.....". Please include figures for this observation to support the claim.

- It is mentioned that "vagal signals are critical for assuring survival after PHx". From the data, it seems to be essential rather than critical for survival as not all the PHx-HV-mice are killed in the experiment.

Vagal signals induce activation of the hepatic FoxM1 pathway in the early phase after PHx

- In figure 1c, BrdU is quantified into % changes in PHx-sham as well as PHx-HV and shows that there is low BrdU-positive hepatocyte in Day5 and Day7. As BrdU staining is used to detect proliferating cells, why, in supplemental figure 1g and h, there are increase in FoxM1 mRNA level as well its target genes especially Mki67 which is the proliferation marker?

Vagal signal-induced activation of the hepatic FoxM1 pathway is necessary and sufficient for promoting hepatocyte replication and resultant whole-body survival in the early phase after PHx

- On top of the results produced in fig2c, d and e, experiments with SO-Ad-hFoxM1 and PHx-sham+Ad-hFoxM1 should be included.

- Although FoxM1 expression is elevated in proliferating cells, FoxM1 level should be relatively low in normal mice. In supplement fig 3a, FoxM1 levels are similar to those of Ad-LacZ controls. Why FoxM1 supplement does not increase FoxM1 levels in liver?

Resident macrophages mediate vagal signal-induced hepatocyte FoxM1 activation and proliferation

- For fig 3e and f, it shows that clodronate liposome administration suppresses FoxM1 activation as well as inhibits the increase in BrdU-positive hepatocytes. It will be useful to induce Ad-hFoxM1 together with clodronate liposome administration to observe whether FoxM1 supplement could rescue the inhibition.

Methods –

- Under Surgical procedures, while some experiment 70% PHx were performed immediately after HV. Which experiment are PHx performed 10 days after HV and why?

- The concentrations of the antibodies and centrifuge speed were not stated.

Figure legends –

- Fig1 d, figures show the relative expression of FoxM1 and its target genes. There is no hepatic weight here but found in supplement fig 1 e, please update accordingly.
- Fig2 a, this is not found in the figure, please update accordingly.

Reviewer #3 (Remarks to the Author):

The authors have responded to my comments and I do not have further comments and suggestions

Responses to reviewer 1

1. The histology of focal necrosis after PHx and HV is most consistent with "biliary infarcts", seen after rupture of portal ductules. The authors should mention that as a also a distinct possibility that needs at some future point to be explained.

In accordance with the reviewer's request, we mentioned that future studies are necessary to elucidate the mechanisms by which HV leads to the pathological changes that we observed in the livers of PHx-HV-mice, including involvement of the bile ducts and rupture of portal ductules. This issue is discussed in the revised manuscript (page 14, lines 7 to 10).

Responses to reviewer 2

1. Most of the results regarding FoxM1 and its target genes concentrate in the mRNA expression level in liver. FoxM1, as a transcription factor, has post-transcriptional modification and thus, mRNA level does not give an actual representation of activity. As such, its target genes may not be the direct effects of increased FoxM1 activity in liver. The reviewer would recommend some western blots on the FoxM1 and its target genes.

We performed western blotting of FoxM1 and its downstream proteins such as Cyclin A2, Cdk1 and PLK1 on postoperative day 2. Consistent with the results of gene expressions, protein levels of those molecules were markedly increased and these increases were blocked by HV. These findings of protein expressions clearly support the notion that the hepatic FoxM1 pathway is activated after PHx and that this activation was inhibited by HV. These results are now presented in Figure 1E and described in the Results section of the revised manuscript (page 7, lines 1 to 4).

2. On top of the use of tamoxifen-inducible liver-specific FoxM1 knock out mice (iFoxM1LKO), using FoxM1 selective inhibitor might be more specific as an administration of tamoxifen could potentially induce nonspecific noises that affect the results. With this in concerned, adenoviral FoxM1 supplement + FoxM1 inhibitor could be used to confirm the FoxM1 activity.

In the experiments using iFoxM1LKO mice, we administered tamoxifen to all mice including the controls. We performed the surgery 1 week after the completion of tamoxifen administration. The T_{1/2} of tamoxifen in mice is reportedly 11.9hr (Robinson SP et al. *Drug Metabolism and Disposition* 19(1) 36-43 1991). Therefore, blood tamoxifen levels were likely negligible when the surgery was performed. In fact, marked up-regulations of FoxM1 and its target genes were observed in tamoxifen-pretreated control mice but these up-regulations were markedly blocked by

hepatic FoxM1 deficiency. These results indicate that nonspecific effects exerted by tamoxifen in these experimental settings are unlikely and that hepatic FoxM1 deficiency per se is responsible for the inhibition of liver regeneration in the early phase after partial hepatectomy.

In contrast, systemic administration of FoxM1 inhibitors to mice, as suggested by the reviewer, affects FoxM1 expressed in all tissues and is less specific for hepatic FoxM1 inhibition. Thus, in our review, experiments employing systemic inhibitor administration would be less definitive than those using tissue-specific knockout mice.

3. It is mentioned that “Until day 5, the weights of livers from both PHx-sham- and PHx-HV-mice were remarkably increased, with an edematous appearance.....”. Please include figures for this observation to support the claim.

As the reviewer requested, we included macro images of the livers from PHx-sham-mice on days 2, 5 and 7 after the operations. The liver of PHx-sham-mice on day 5 showed an edematous and whitish appearance, as compared to the livers on day 7. These results are now presented in Supplemental Figure S1C of the revised manuscript.

4. It is mentioned that “vagal signals are critical for assuring survival after PHx”. From the data, it seems to be essential rather than critical for survival as not all the PHx-HV-mice are killed in the experiment.

We agree with the reviewer’s comment. We changed “critical” to “essential” in the Results section (page 5, line 2 and page 5, line 17).

5. In figure 1c, BrdU is quantified into % changes in PHx-sham as well as PHx-HV and shows that there is low BrdU-positive hepatocyte in Day5 and Day7. As BrdU staining is used to detect proliferating cells, why, in supplemental figure 1g and h, there are increase in FoxM1 mRNA level as well its target genes especially Mki67 which is the proliferation marker?

As shown in Figure 1C, BrdU-positive cell rates of PHx-sham mice were still significantly higher than those of SO-mice on days 5 and 7, although the magnitudes were much smaller than that on day 2. Therefore, these BrdU incorporation results are consistent with those indicating that expressions of FoxM1, its target genes and *Mki67* were increased in the livers of PHx-sham mice on days 5 and 7, as shown in Supplemental Figure S1G and S1H of the original manuscript (Figure S1H and S1I of the revised manuscript). As the reviewer suggested, however, the data obtained from gene expression analysis and the BrdU assay were quantitatively different, possibly due

to the following numerous factors. First, the BrdU assay marks proliferating cells at the S phase of the cell cycle, whereas Mki67 is a marker of cell proliferation at the M phase. Second, the BrdU assay detects the number of proliferating cells, while gene expression analysis shows the mRNA amounts of each gene in whole liver cells. Third, gene expression analysis is very sensitive, because gene expressions are detected employing the PCR method. On the other hand, the BrdU assay includes several BrdU incorporation processes *in vivo* and immunohistochemistry, which may reduce the sensitivity of detection. These factors may account for the quantitative differences between FoxM1-related gene expressions and the numbers of BrdU-positive cells. We added a brief explanation focusing on this point to the Results section (page 7, lines 8 to 9).

6. On top of the results produced in fig2c, d and e, experiments with SO-Ad-hFoxM1 and PHx-sham+Ad-hFoxM1 should be included.

As the reviewer suggested, we examined the effects of adenoviral FoxM1 expression in the liver on the hepatocyte proliferation responses in SO- and PHx-mice. Consistent with the data shown in Supplemental Figure S3B, exogenous FoxM1 expression in SO-mice did not significantly affect either the expressions of cell cycle-related genes or the numbers of BrdU-positive hepatocytes. In addition, adenoviral FoxM1 overexpression in the livers of PHx-sham-mice did not yield additional increases in the expressions of cell cycle-related genes or in the numbers of BrdU-positive hepatocytes. These findings suggest that the hepatic FoxM1 pathway had already been fully activated in response to PHx. These results are now presented in Supplemental Figures S3C and S3D and described in the Results section of the revised manuscript (page 8, lines 15 to 18).

7. Although FoxM1 expression is elevated in proliferating cells, FoxM1 level should be relatively low in normal mice. In supplement fig 3a, FoxM1 levels are similar to those of Ad-LacZ controls. Why FoxM1 supplement does not increase FoxM1 levels in liver?

As described in the original manuscript, we used adenovirus containing the human *Foxm1* gene to distinguish endogenous FoxM1 from exogenously expressed FoxM1. Supplemental Figure S3B shows the hepatic expression of murine endogenous FoxM1 after adenoviral supplementation of human FoxM1. The results obtained indicate that exogenous FoxM1 supplementation did not affect endogenous FoxM1 expression. To avoid the misunderstanding, we now clearly describe that expressed protein was the

human FoxM1 (page 8, line 12).

8. For fig 3e and f, it shows that clodronate liposome administration suppresses FoxM1 activation as well as inhibits the increase in BrdU-positive hepatocytes. It will be useful to induce Ad-hFoxM1 together with clodronate liposome administration to observe whether FoxM1 supplement could rescue the inhibition.

As suggested, we performed FoxM1 supplementation in clodronate liposome-treated mice and examined the acute hepatocyte proliferation responses of these mice. As expected, hepatic FoxM1 supplementation significantly blunted the inhibitory effects of macrophage depletion on post-PHx increases in hepatic expressions of FoxM1 and its target genes as well as in BrdU-positive hepatocytes. These results further strengthen our conclusion that hepatic macrophages contribute to mediating vagal signals to hepatocytes, thereby activating the hepatocyte FoxM1 pathway and promoting hepatocyte proliferation. These results are now presented in Supplemental Figures S4B and S4C and are also described in the Results section of the revised manuscript (page 10, lines 3 to 6).

9. Under Surgical procedures, while some experiment 70% PHx were performed immediately after HV. Which experiment are PHx performed 10 days after HV and why?

In the first revision process, one of the reviewers raised the possibility that unexpected effects caused by vagotomy immediately before PHx were involved in hepatocyte proliferation after PHx. Therefore, we performed vagotomy 10 days before PHx and examined post-PHx hepatocyte proliferation. As shown in Supplemental Figure S1C of the original manuscript (Supplemental Figure 1D of the revised manuscript), vagotomy 10 days before PHx similarly blocked hepatocyte proliferation. Based on these results, we showed the data, obtained from experiments in which vagotomy was performed immediately before PHx, in all other related figures. We now clearly describe this issue in the Methods section (page 20, line 6).

10. The concentrations of the antibodies and centrifuge speed were not stated.

As suggested, we added this information to the Methods section (page 23, lines 20 to 22).

11. Fig1 d, figures show the relative expression of FoxM1 and its target genes. There is no hepatic weight here but found in supplement fig 1 e, please update accordingly.

We apologize for the erroneous description in the figure legends. We corrected the

legend of Figure 1D of the revised manuscript.

12. Fig2 a, this is not found in the figure, please update accordingly.

We apologize for the erroneous description of the figure legend. We removed the legend for Figure 2A and corrected the labeling of the other figure legends in Figure 2 of the revised manuscript.

REVIEWERS' COMMENTS:

Reviewer #2 (Remarks to the Author):

While the reviewer would like to see more novel mechanistic data on how hepatic FoxM1 is activation, one also feels that the authors have made satisfactory efforts within their ability to answer the reviewers' critiques. With this in mind, the reviewer would tend to agree with acceptance for publication in the present form.